# Design of Wide Particle Size Range Aerodynamic Injection System with New Pre-focus Structure

**Junhong Huang** [1]**, Lei Li** [2,3]**, Xue Li** [2,3]**, Zhengxu Huang** [2,3] **and Zhi Cheng**[4]

[1]Guangdong MS institute of scientific instrument innovation, Guangzhou, 510632, China

[2]Institute of Mass Spectrometry and Atmospheric Environment, Jinan University, Guangzhou, 510632, China

[3]Guangdong Provincial Engineering Research Center for On-Line Source Apportionment System of Air Pollution, Guangzhou, 510632, China

[4]Institute of Systems Engineering, Academy of Military Sciences, Tianjin, 300161, China

**Correspondence:** Lei Li (lileishdx@163.com)

**Abstract:** A new aerodynamic injection system has been designed for wide particle size range, combining a new pre-focus structure, a smaller buffer chamber and a five-stage lens. Compared with previous injection systems, the new design adds virtual impactors and pre-focus structures, while reducing the overall volume by 90 %. The newly developed pre-focus structure effectively addresses the challenges associated with the focusing and transmission of large particles, significantly reducing the beam width and dispersion angle of particles exiting the critical hole, thus preventing the buffer chamber from becoming excessively large. Furthermore, the focusing capability and transmission efficiency for large particles have been significantly enhanced, with the transmission range expanded to encompass particles from 100 nm to 10 μm. Numerical simulations demonstrate that the injection system can transmit particles within the 1 to 9 μm range with an efficiency exceeding 90 %. Additionally, the standard microsphere experiment verified the good consistency between the performance of the injection system and the simulation results. In the testing of standard dust, the wide-range particle size distribution obtained by the new injection system is highly consistent with the Aerodynamic Particle Sizer (APS). In summary, this new design has ultra-high transmission efficiency while reducing volume, demonstrating the miniaturization potential of single particle aerosol mass spectrometer in detecting particles with a wide particle size range.

**Keywords:** Aerosol particles; Aerodynamic lens; Beam width; Transmission efficiency; Numerical simulation

## 1 Introduction

As a key component of aerosol mass spectrometry, the particle beam generator is used to focus the injected particles, and the focusing ability of the particle beam determines the detection sensitivity of the aerosol mass spectrometer. Aerodynamic lenses, which are recognized as a widely utilized particle beam focusing technology (Liu et al., 1995; Murphy et al., 2006; Zelenyuk et al., 2015; Clemen et al., 2020), leverage the inertia differential between particles and surrounding fluids to effectively focus particles, finding extensive application across various aerosol mass spectrometry systems (Peck et al., 2016). In addition to its application in aerosol mass spectrometry, aerodynamic lens systems have also found applications in many analytical instruments. Researchers use aerodynamic lenses to introduce aerosol particles into pulsed X-ray beams and determine the particle composition using diffraction patterns and ion fragments generated when the X-ray pulse meets the particle (Loh et al., 2012). Aerodynamic lenses can also be used in the mass spectrometry of nano-mechanical resonators. The lenses enable efficient focusing and transmission of large analytes without ionization, thus significantly enhancing overall system performance (Dominguez-Medina et al., 2018).

While aerodynamic lenses demonstrate a significant transmission effect on particles, most current designs primarily focus on a particle size range that falls within the same order of magnitude, with effective focusing predominantly limited to particles smaller than 3 μm. (Fergenson et al., 2004; Srivastava et al., 2005; Tobias et al., 2006). We typically assess the particle transmission capacity of injection systems by considering the range where the transmission efficiency exceeds 50 %, such as the 25-250 nm of Liu et al. (1995), the 100-900 nm and the 340-4000 nm of Schreiner et al. (1998; 1999), the 60-600 nm of Zhang et al. (2004), and the 125-600 nm of Zelenyuk et al. (2015). The size of atmospheric aerosols spans from sub-nanometer to millimeter scales. Consequently, broadening the particle focusing range is essential for enhancing the analytical capabilities of aerosol mass spectrometry, particularly in the examination of biological aerosols, dust, and single cells.

Researchers have been actively working to extend the particle transmission range by optimizing aerodynamic lenses. Research has found that the focusing performance of aerodynamic lenses for small particles below 50 nm is limited by Brownian motion, while focusing of large particles is mainly affected by the larger inertia of particles (Wang et al., 2005; Wang and McMurry, 2006).

At present, most reported lenses exhibit inefficient transport of particles larger than 5 μm(Cahill et al., 2014; Deng et al., 2008; Williams et al., 2013; Wu et al., 2009). The significant expansion of particle transport range relies on the improvement of the transport performance of large particles, and the most direct way to achieve this goal is to increase the number of lens stages. For example, Lee et al.(2013) designed a seven-stage lens for particle detection in the range of 30 nm to 10 μm, but this study does not consider the impact of critical hole on the transmission loss of large particles and it is not applied in practice. Cahill et al. (2014) designed a high-pressure lens,

and used a very long buffer chamber combined with a seven-stage lens to transport 4-10 μm
particles. However, the transmission efficiency of 4 μm and 9 μm particles in the experiment was
only 20 %, and the overall size of the lens system was relatively large. Increasing the number of
lens stages is obviously beneficial for expanding the transmission range of particles, but it also
brings disadvantages such as excessive injection system volume and increased assembly difficulty.
As mentioned by Liu et al. (2007) in their report, the beam focusing effect decreased as the
assembly accuracy of the lens system decreased.
Hari et al. (2007) added a virtual impactor behind the critical hole to reduce cross trajectory
phenomenon of large particles, which was beneficial for enhancing the transmission performance
within the injection system. Chen et al. (2007) studied the effect of the structural configurations
before and after the critical hole on particle loss through numerical simulation and experimental
verification. They improved the transport efficiency of particles that are 50 nm or smaller by
optimizing the structure. However, the transport range did not significantly expand. Liu et al.
(2007) reduced the wall impact loss of particles by unifying the inner diameter of the downstream
pipeline of the critical hole. Additionally, Hwang et al. (2015) proposed a novel convergent-
divergent critical hole that effectively reduces the incident angle of large particle and transports
particles in the range of 30 nm to 10 μm efficiently. However, due to technological limitations, this
new type of critical hole is difficult to manufacture and has not yet been applied in practice. In
conclusion, the above research indicated that large particles transmission in aerodynamic lens
systems experience significant losses on the surface of the critical holes.
Du et al. (2023) first proposed a pre-focus design that includes a two-stage lens positioned at
the front of the critical hole. With the aid of this pre-focus structure, particles are effectively
aligned along the axis before entering the critical hole. This design significantly reduces the
incidence angle of large particles, thereby minimizing the impact loss on the surface of the critical
hole. However, due to the large divergence angle of particles after they exit the critical hole, it is
essential to incorporate a buffer chamber with a diameter of 250 millimeters, along with a seven-
stage lens system, to efficiently transport particles ranging from 62 nm to 13 μm. Nevertheless,
this design markedly increases the overall size of the injection system, which poses challenges for
the miniaturization of the mass spectrometer.
This study designs a new small volume pre-focus wide range aerodynamic lens injection
system (PFW-ALens) for particles with diameters ranging from 100 nm to 10 μm. The injection
system consists of a novel pre-focus structure, a small buffer chamber, and a five-stage
aerodynamic lens. This study uses computational fluid dynamics software to simulate the
transmission performance from the inlet to the vacuum chamber, and verifies it through
experiments. The numerical simulation results demonstrate that the particle transmission
efficiency is greater than 90 % for sizes ranging from 100 nm to 9 μm. The results obtained from
the experiments demonstrate significant consistency with the simulation outcomes, particularly
revealing that efficiency exceeds 50 % for particles up to 9 μm. This injection system has great

potential for application in bioaerosol analysis, dust analysis, and miniaturization of mass spectrometers.

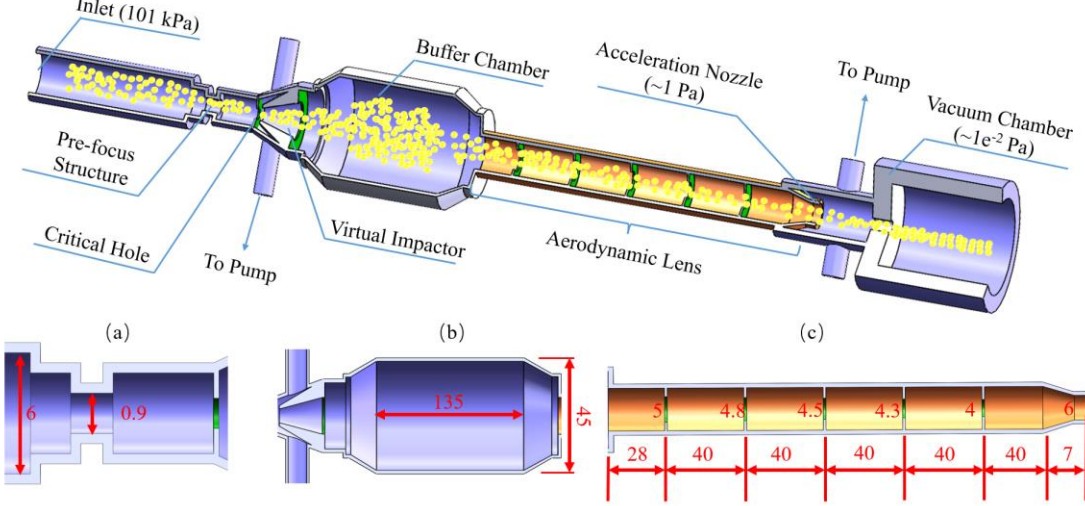

Fig. 1 Structural design of the injection system. Locally displayed are the pre-focus structure (a), buffer chamber (b), and five-stage lens system (c).

## 2 Numerical simulation and experimental design

### 2.1 Physical models

Fig. 1 shows the structure and key dimensions of the injection system, which consists of five modules: pre-focus hole, the critical hole, the virtual impactor, the buffer chamber, and a five-stage aerodynamic lens. The distance from the inlet of the buffer chamber to the outlet of the acceleration nozzle in this study is 370 mm, compared to 570 mm in the injection system designed by Du et al. The particles are first focused near the axis with the aid of the pre-focus structure before passing through the critical hole, which has a diameter of 0.26 mm, at a flow rate of 640 mL/min. A virtual impactor, located 1.6 mm downstream of the critical hole, features a diameter of 1 mm and an angle of 30 °. In this study, the virtual impactor is defined as a device that enhances the concentration of larger particles in the sample air by employing a pressure-reduction orifice and transverse pumpout design. The addition of a separation cone further refines the virtual impaction effect, distinguishing it from the inherent virtual impaction observed in other aerosol mass spectrometer systems referenced in the literature. The excess air between the critical hole and the virtual impactor is extracted by a vacuum pump to condense the aerosol particles entering the buffer chamber. As the high-speed particles pass through the critical hole, they gradually decelerate within the buffer chamber. The airflow then drives them into the aerodynamic lens system, which composed of holes with different diameters that effectively focus particles onto the central axis within a specific range. In addition, this study utilized a tapered nozzle at the end of the aerodynamic lens. As mentioned by Zhang et al. (2004) in their study, this nozzle provides better collimation for small particles and improves the transport efficiency of large particles

compared to stepped nozzles. Downstream of the nozzle, a second pumping stage is employed to
further reduce the pressure and enhance particle acceleration into the vacuum chamber. This stage
utilizes a molecular pump to achieve the necessary pressure drop, ensuring efficient particle
transport and stable operation. Both pumping lines are maintained at a constant pressure using
commercial pressure controllers, which provide precise control of the flow dynamics. The
particles focused by the lens group will be further accelerated into the vacuum chamber by the
nozzle.
**2.2 Numerical model**
Ansys-Meshing is used to generate the mesh, and Fluent software is used to calculate the
flow of gas and particle coupling between the inlet and vacuum chamber. The boundary conditions
for the inlet of the injection system, the pumping port of the virtual impactor, and the outlet of the
lens are set to 101325 Pa, 600 Pa, and 0.1 Pa, respectively. Under these conditions, the buffer
chamber and the aerodynamic lens operate below 300 Pa.
The numerical simulation employs an ideal gas as the material property, which means it
considers the characteristics of compressible flow, particularly as the gas undergoes rapid
expansion and acceleration downstream of the critical hole. In this region, the compressible
Navier-Stokes equations are used to resolve pressure gradients, density variations, and inertia
effects, all of which are essential for capturing transitions between supersonic and subsonic flow,
as well as shock phenomena. To reduce computational complexity, axial symmetry is rigorously
applied, assuming that flow properties and particle trajectories are symmetric around the central
axis of the hole.
In addition, a laminar flow model is employed for simulation, and the second-order upwind
flow equation is selected as the discretization scheme. The parameter settings are consistent with
the study of Zhang et al. (2002). During the simulation, user-defined functions (UDFs) are utilized
to modify and implement particle drag and Brownian force models, allowing for more accurate
particle motion trajectories. Finally, the discrete phase model (DPM) is employed to define the
number, diameter, and release position of the particles. Upon completion of the simulation, the
trajectories and velocities of the particles are obtained through contours.
**2.3 Experimental exploration**

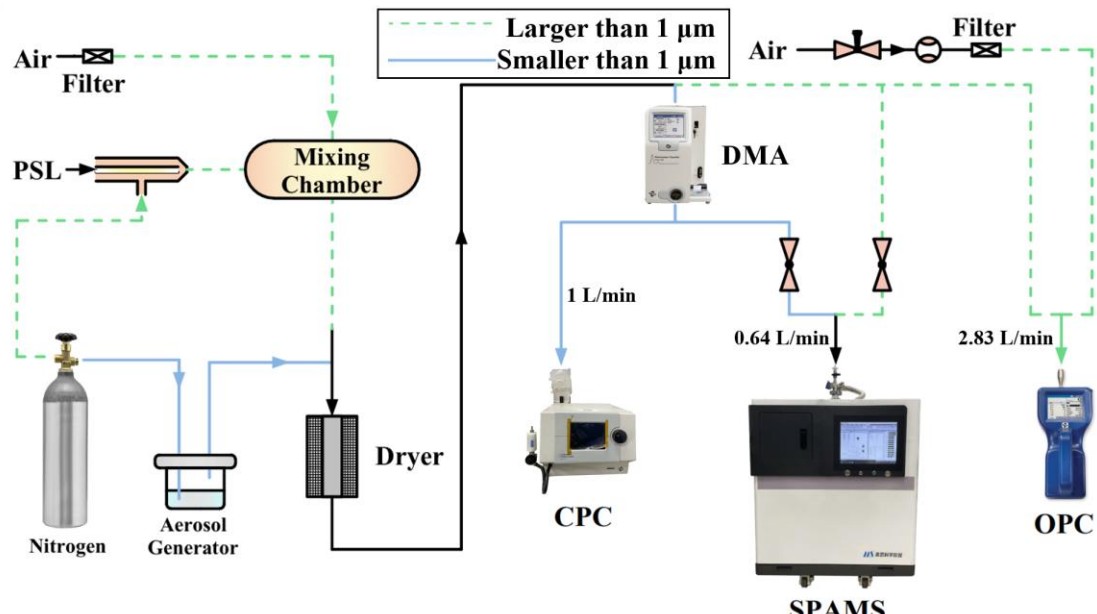

Fig. 2 Gas Path Connections and Flow Rates (L/min) to Each Detector in Experiment

Standard polystyrene latex spheres (PSL, Thermo Fisher Scientific) ranging from 100 nm to 10 μm are used to characterize the focusing ability of aerodynamic lens system. The operation for generating and counting aerosols larger than 1 μm is as follows. Initially, the PSL solution is diluted with pure water, after which nitrogen serves as the carrier gas to atomize the PSL solution using an ICPMS atomizer (Ge, C21-1-UFT02). The atomized mixture is subsequently directed into the mixing chamber, where the injection rates of the PSL solution and nitrogen are meticulously set to 10 μL/min and 0.2 L/min, respectively. The excess moisture in the atomized aerosol particles is removed by heating the mixing chamber and introducing a drying tube, and then the aerosol particles are introduced into the optical particle counter (OPC, TSI, Model 9306) and Bio-SPAMS respectively through a three-way flow splitter for counting. For particles smaller than 1 μm, an aerosol generator (TSI, Model 9302) is used to produce aerosols. The generated aerosols are first sorted by a differential mobility analyzer (DMA, TSI, Model 3082) and then directed to a condensation particle counter (CPC, TSI, Model 3775) and biological aerosol single particle mass spectrometer (Bio-SPAMS) for counting respectively. The detailed experimental gas path connection is shown in Fig. 2, where the blue solid line is the test pipeline connection scheme for the transmission efficiency of particles below 1 μm, and the green dotted line represents the test pipeline connection scheme for the transmission efficiency of particles above 1 μm. It is important to note that during the experiment, we utilized additional airflow. For experiments involving particles smaller than 1 μm, this additional airflow was implemented to address flow mismatches between the aerosol generator and the detection instruments. In experiments with particles larger than 1 μm, the additional airflow helped resolve the issue of unequal flow rates between the OPC and Bio-SPAMS, ensuring that the flow rates entering both instruments remained consistent during measurement.

The Bio-SPAMS used in this study is similar to HP-SPAMS (Du et al., 2024), but its optical

system has been improved. The first diameter measuring laser is designed with a beam-splitting optical structure similar to that of the Aerodynamic Particle Sizer (APS, TSI, Model 3321). The method involves using a beam splitter to divide the diameter measuring laser (Sony SLD3234VF) into two nearly parallel beams and calculating the aerodynamic diameter of the particles by the time they pass through these beams. Additionally, Bio-SPAMS introduces a laser-induced biofluorescence detection module, enabling the pre-screening of fluorescent particles. This allows for selective ionization of only the fluorescent particles, meeting the requirements for online monitoring of bioaerosols.

The photomultiplier tube detects the number of particles passing through the laser by collecting light signals, and PMT 1-1 is used to count the number of particles passing through the first laser of the split beam. In the experiment, we defined the total number of particles detected by PMT 1-1 per unit time as the total number of particles entering Bio-SPAMS. For particles larger than 1 μm, the ratio of the total number of particles entering both Bio-SPAMS and OPC is used to determine the transmission efficiency. For particles smaller than 1 μm, the transmission efficiency is calculated as the ratio of the particle concentration recorded by Bio-SPAMS to that recorded by CPC.

## 3 Results and discussions

### 3.1 Injection system with virtual impact

Currently, the lens designed by Zhang et al. (2004), has been applied by numerous researchers (Canagaratna et al., 2007; Docherty et al., 2013; Drewnick et al., 2009; Meinen et al., 2010) across various fields. However, the simulation results presented by Zhang et al. indicate that the focusing range of such lenses primarily lies between 50 nm and 3 μm, with a transmission efficiency of 50 % at 1.5 μm, as illustrated in Fig. 3(a). Furthermore, Zhang et al. assumed in their simulations that particles were uniformly distributed from the buffer chamber to the area in front of the lens. In reality, however, the entry conditions for larger particles are not uniform. To gain further insights into the impact of this lens on particles within mass spectrometry instruments, this study enhances Zhang et al.'s model by incorporating a buffer chamber and a virtual impactor structure, as depicted in Fig. 1(b), and conducts relevant simulations for exploration.

Our team discovered that the virtual impactor used in this study is capable of transporting 100 nm particles downstream with an efficiency of over 90 %, with only a small fraction of particles being pumped away. This indicates that nearly all of the particles examined in this research can pass through the virtual impactor and be effectively transported downstream. In order to further compare the effects of the virtual impactor and pre-focus structure with the five-stage lens employed in this study, this study first removed the virtual impactor and pre-focus structure from Fig. 1 and simulated the transmission efficiency of the model (represented by the blue left triangle line). Subsequently, the virtual impactor (orange diamond line, original design) and the pre-focus structure (black square line, present design) were sequentially reintroduced to observe

the enhancements in transmission efficiency. By comparing the transmission effects of three
design above, the advantages of the design in Fig. 1 are highlighted.
The transmission efficiency presented in this study is the ratio of the number of particles at a
distance of 110 mm from the lens outlet to the number of particles at the inlet of the sampling
system. The beam width of the particles is determined by selecting the radial distribution of 90 %
of the particles at this same distance. The reason for choosing the 110 mm position is that it is
downstream of all lasers, including the acceleration nozzle to PMT1 (42 mm) and PMT2 (67 mm),
allowing for a clearer evaluation of the particle transport and focus effect.
As shown in Fig. 3, by comparing the particle transmission efficiency curves before and after
adding a virtual impactor, it can be found that after the addition of the virtual impactor (orange
diamond symbol line), the focusing ability of the injection system for particles larger than 1 μm
has increased to varying degrees, and the transmission efficiency of 7 μm particles has increased
from the initial 5 % to 30 %. This improvement is primarily attributed to the virtual impactor,
which refines the flow dynamics, reduces particle loss, and optimizes the impaction process,
thereby enhancing the focusing efficiency for larger particles. Although the aerodynamic lens
system with virtual impactor has increased its ability to focus on large particles, this particle size
range is still insufficient for the detection of large particle such as dust and organisms.

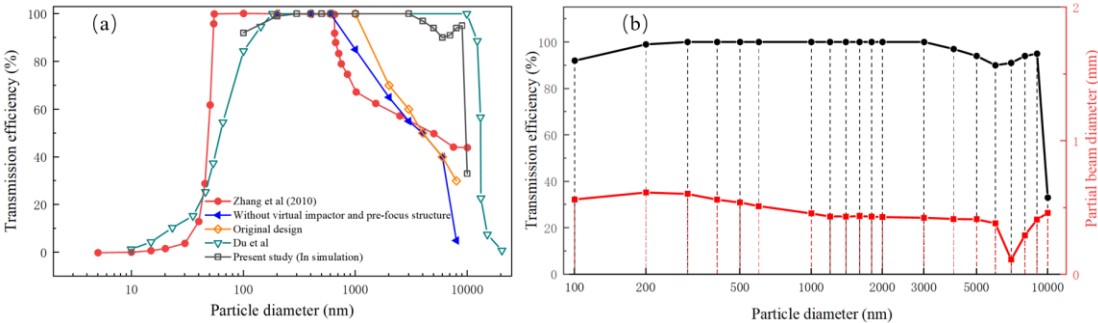


Fig. 3 Performance of different injection systems in simulation. Fig. 3 (a) shows the particle
focusing range of different injection systems, and Fig.3 (b) shows the transmission efficiency and
particle beam width of the injection system in this study (PFW-ALens) in terms of simulation.
**3.2 Injection system with pre-focus structure**
Du et al. proposed a pre-focusing technique that incorporates a set of two-stage lenses in
front of the critical hole to address the low efficiency of transporting larger particles mentioned in
section 3.1. From Fig. 3(a), it can be observed that the pre-focus injection system designed by Du
et al. can reduce the loss of 10 μm particles from about 40 % to 0 %, achieving efficient transport
within the range of 0.18-10 μm. Although Du et al.'s injection system can cover a wide range of
particle sizes, there are still some drawbacks to the overall system. The main problem is that the
particle beam diverges significantly after passing through the critical hole and the virtual impact
hole. Therefore, a buffer with a diameter of 250 millimeters and a height of 250 millimeters is
needed to accommodate all particles in the study.

Fig. 4 compares the radial width of the particle beams for 0.5 μm, 1 μm, 3 μm, and 5 μm particles at different positions in the original design, Du et al.'s design, and the design proposed in this study. Compared to the original design, the pre-focus structures designed by Du et al. and the one proposed in this study both reduce the radial width of the particle beams at the critical hole, the inlet of the virtual impactor, and the outlet of the nozzle. It is worth noting that the pre-focus structure designed by Du et al. significantly increases the radial distribution of the particle beam in the buffer cavity when transmitting large particles such as 3 μm (fig. 4(c)) and 5 μm (fig. 4(d)). This poses higher requirements for the focusing of the aerodynamic lenses located behind the buffer chamber, and a common approach to achieve this is to use additional stages of lenses for improved focusing. When combined with a seven-stage aerodynamic lens, the overall dimensions and volume of the injection system can reach a diameter of 278 millimeters and a height of 875 millimeters. A large-sized injection system is not only detrimental to the miniaturization of mass spectrometers, but the seven-stage lens can also lead to increased assembly complexity, making it difficult to ensure the consistency of the lenses.

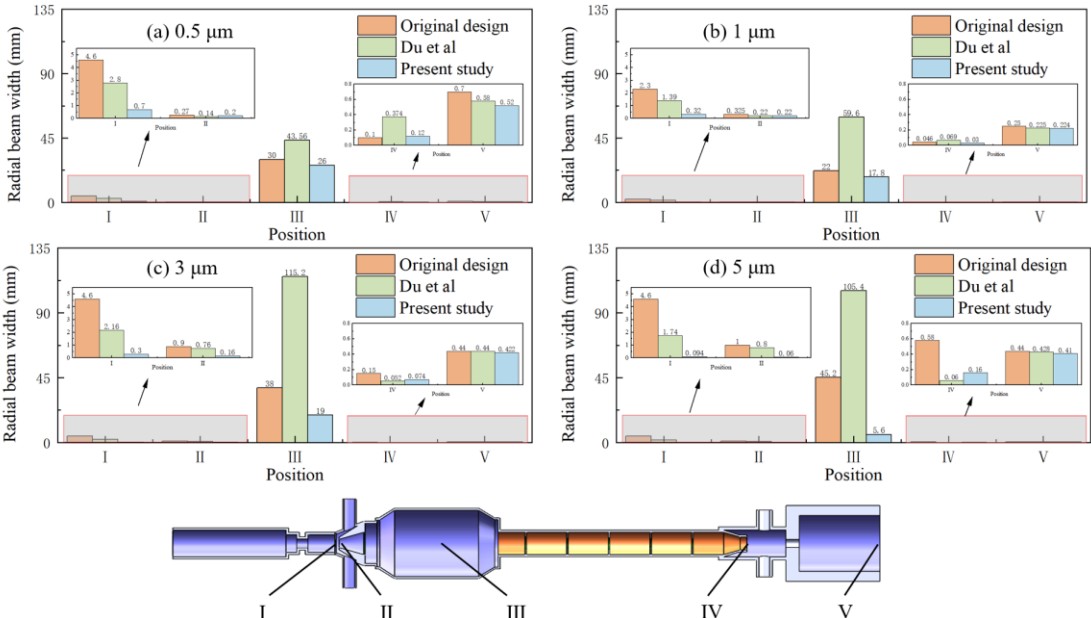

Fig. 4 Comparison of the radial width of the particle beam in the key components of the original design, Du et al.'s design, and the pre-focus injection system proposed herein. The particles characterized are 0.5 μm, 1 μm, 3 μm, and 5 μm, as represented by (a) to (d), respectively. Specifically, (I) denotes the front end of the critical hole, (II) refers to the inlet of the virtual impactor, (III) designates the buffer chamber, (IV) indicates the outlet of the acceleration nozzle, and (V) represents the outlet of the vacuum chamber.

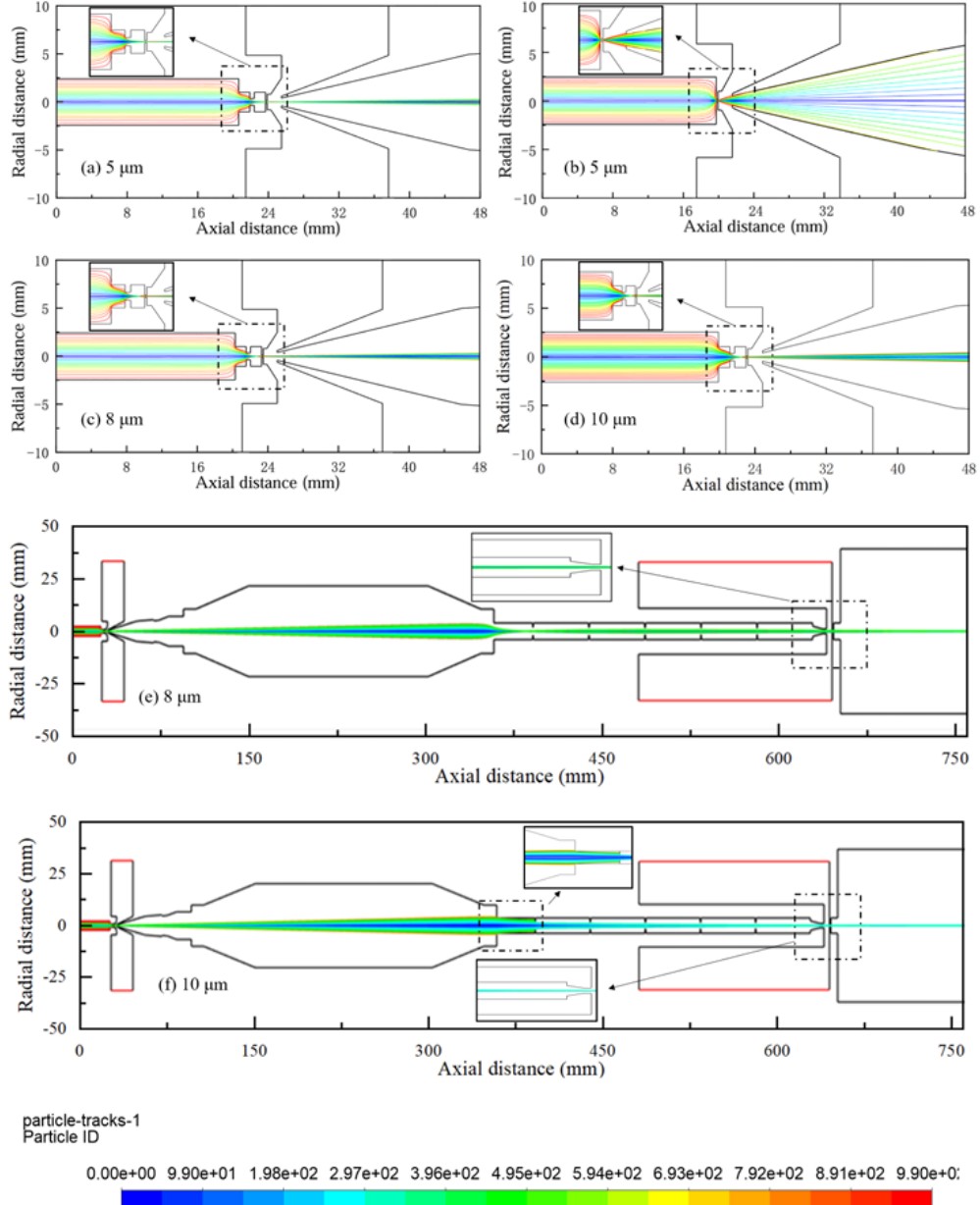

Fig. 5 The trajectories of large particles in various injection systems are as follows: (a) shows the trajectory of 5 μm particles in the current design; (b) depicts the trajectory of 5 μm particles in the original design; (c) illustrates the trajectory of 8 μm particles in the current design; (d) presents the trajectory of 10 μm particles in the current design; (e) and (f) display the trajectory distributions of 8 μm and 10 μm particles within the overall sampling system, respectively.

**3.3 New design**

Section 3.2 investigates the impact of the pre-focus structure proposed by Du et al. on the radial width of the particle beam, which directly influences the buffer chamber dimensions and particle focusing efficiency. To address these issues, this study proposes a single-stage lens pre-focus structure, with detailed dimensions provided in Fig. 1.

As shown in Fig. 4, the PFW-ALens-equipped injection system significantly reduces the

radial width of the particle beam at various positions across different particle diameters, outperforming both the original design and Du et al.'s pre-focusing structure. Specifically, at the buffer chamber, the beam width is reduced by 70 % to 95 % compared to Du et al.'s design. Additionally, the radial distribution of particles in the buffer chamber exhibits an inverse correlation with particle size, a novel observation not seen in previous pre-focusing designs. This discovery offers a strong basis for optimizing the lens system's length and minimizing the number of lenses required.

The divergence angle of the beam after particles pass through the critical orifice directly influences the design of the buffer chamber and the transmission efficiency of the particles (Lee et al.,2013). A greater degree of divergence increases the likelihood of particles colliding with the walls, leading to losses, which has been a limitation in previous injection system designs. Fig. 5 illustrates the advantages of the proposed PFW-ALens in transmitting larger particles. Fig. 5(a) and 5(b) depict the transmission trajectories of 5 μm particles in the PFW-ALens and the original design system, respectively. It can be observed that in the original design, the divergence significantly increases after the particles pass through the critical orifice, resulting in collisions with the virtual impactor and subsequent losses. Conversely, Fig. 5(c) and 5(d) show the trajectories of larger particles (such as 8 μm and 10 μm) in the PFW-ALens, revealing that the divergence angle after leaving the critical orifice is markedly small, leading to a narrow beam width throughout the entire system. This is in stark contrast to the results obtained using the pre-focusing structures designed by Du et al. Furthermore, the dimensions of the buffer chamber used in this study are 45×135 mm, which represents over a 90 % reduction in volume compared to the buffer chamber designed by Du et al.

Fig. 3(b) demonstrates a remarkable performance in the particle transmission range of the PFW-ALens. Within the particle size range of 100 nm to 9 μm, the transmission efficiency remains above 90 %, with a notable decline in efficiency only observed for 10 μm particles. Furthermore, simulation results indicate that the beam width can be maintained at less than 0.6 mm across the entire range of particle sizes. Furthermore, simulation results indicate that the beam width can be maintained at less than 0.6 mm across the entire range of particle sizes. Notably, the laser employed in this study differs from that utilized by Du et al.; the former has a laser beam diameter of 3 mm. The new optical system adjusts the beam to a rectangular spot of 600 × 30 μm using shaping optical elements, which means the adjusted spot can essentially cover the beam width for different particle sizes, providing favorable conditions for experimental testing. However, simulations also reveal that design improvements, such as increasing the length of the buffer chamber, can further enhance the transmission efficiency for 10 μm particles, indicating that the effective transmission of 9 μm particles does not represent the limit of this system. Overall, the particle focusing capability of the PFW-ALens for particles in the 10 μm range is nearly comparable to the focusing performance observed by Du et al. within the same transmission range. It's noteworthy that this study employs a five-stage lens, in contrast to the seven-stage lens

commonly used in most wide-range injection systems. This suggests that the pre-focus device utilized in this research not only achieves more effective focusing of larger particles but also does so with fewer lenses, thereby reducing the overall volume of the injection system. This advancement is of significant importance for the miniaturization of single-particle mass spectrometers.

**3.4 Validation and application**

Although a beam width of 600 μm was employed in this study, the particle beam width for certain small particles still exceeds this threshold. According to Rayleigh scattering theory, the scattered light intensity is proportional to the sixth power of the particle diameter, leading to a dramatic decline in signal intensity as the particle size decreases. Due to the combined effects of these two factors, the detection sensitivity and transmission efficiency of small particles (e.g., 100 nm) are significantly compromised, as indicated by the low transmission efficiency of 100 nm particles in Fig. 6.

Fig. 6 compares the transmission efficiency of particles with different particle sizes in numerical simulation and experimental validation of PFW-ALens. The results indicate a strong correlation between the experimental findings and the numerical predictions. Error bars represent the range between the maximum and minimum values from five independent experimental replicates, with the midpoint denoting the average value. Within the particle size range of 200 nm to 6 μm, the measured transmission efficiency remains above 90 %. However, for 100 nm particles, the experimental results reveal substantial losses. This discrepancy is primarily due to the use of a dual-peak signal by the Bio-SPAMS system for particle counting; to accommodate the requirements of biological fluorescence detection, the intensity of the second beam must be reduced, resulting in lower light scattering intensity for 100 nm particles, which in turn hinders accurate counting. For particles larger than 6 μm, the testing results show a marked decrease in transmission efficiency. Nevertheless, 9 μm particles can still maintain an efficiency of 65 %. The trend in transmission efficiency for 10 μm particles is consistent with the simulation results, registering only 25 % efficiency. The specific transmission efficiencies for different particle sizes are summarized in Table 1.

Table 1. Comparison of Transmission Efficiency Across Different Particle Sizes

| Particle diameter (nm) | Transmission efficiency (%) | Particle diameter (nm) | Transmission efficiency (%) |
|---|---|---|---|
| 100 | 35 | 4000 | 95 |
| 200 | 100 | 5000 | 91.65 |
| 300 | 107 | 6000 | 88.3 |
| 500 | 104 | 7000 | 72.1 |
| 800 | 101 | 8000 | 65.7 |
| 1000 | 108.5 | 9000 | 64.9 |
| 2000 | 101 | 10000 | 25 |
| 3000 | 107.5 | | |

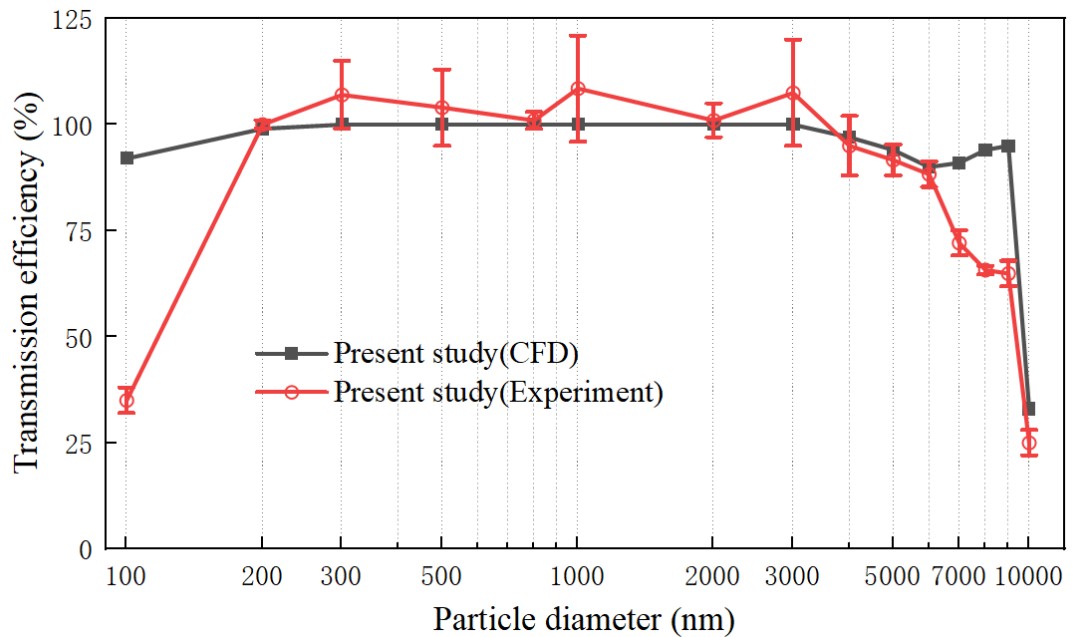

Fig. 6 Experimental verification of particle transmission

To characterize the analytical capabilities of the PFW-ALens for large particles, standard ultrafine dust (ISO 12103, PTI) was selected as the test sample. Initially, APS was employed as the standard to detect the dust. Subsequently, tests were conducted and compared using the PFW-ALens (the red dashed line) and the original system (the black solid line), with the results shown in Fig. 7. The specific particle size distribution is illustrated in the same figure, where the Y-axis represents the normalized dN/dlogD.

The APS measures the aerodynamic diameter ($d_{ca}$) in the continuum flow regime, while the Bio-SPAMS determines the vacuum aerodynamic diameter ($d_{va}$) under vacuum or near-vacuum conditions. Since the tested samples were high-density non-spherical particles (with particle density $\rho_p > \rho_0$, where $\rho_0 = 1 \mathrm{g/cm^3}$ is the standard reference density), the theoretical ratio between the two aerodynamic diameters can be expressed as:

$$\frac{d_{va}}{d_{ca}} = \sqrt{\frac{\rho_p \chi_c}{\rho_0 (\chi_v)^2}}$$

where $\chi_c$ and $\chi_v$ ($\geq 1$) are dynamic shape factors correcting aerodynamic drag in the continuum (viscous-dominated) and free-molecular (collision-dominated) regimes, respectively. Since $\rho_p > \rho_0$ and $\chi_c/(\chi_v{}^2) > 1$ for such particles, $d_{va}$ exceeds $d_{ca}$, resulting in a rightward shift of the Bio-SPAMS distribution curve relative to APS in Fig. 7, where X-axis represents aerodynamic diameter.

It is evident that the particle size distribution obtained using the PFW-ALens closely aligns with that measured by the APS. The above experiments not only prove that PFW-ALens can more accurately detect the particle size distribution of dust compared to injection systems without pre-focus holes, but also prove that PFW-ALens can achieve efficient transmission of large particles, demonstrating the potential of new lens structure in detecting large particle analysis.

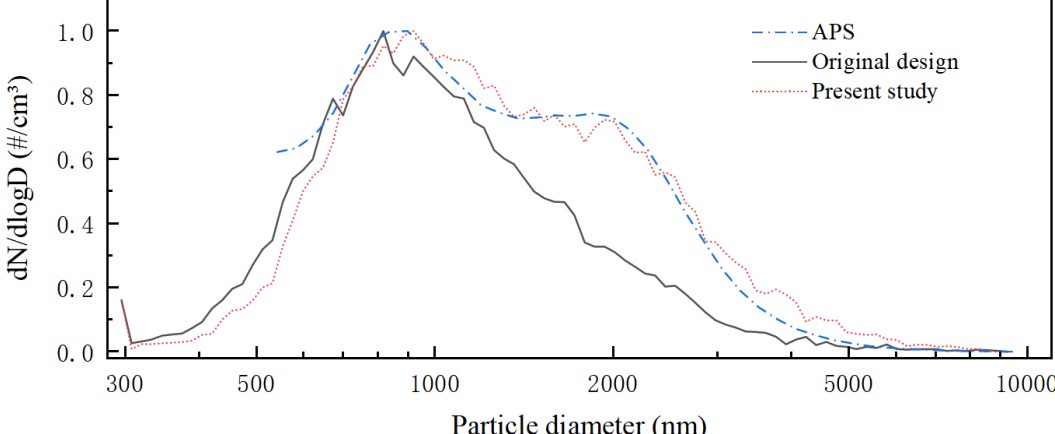

Fig. 7 3-Minutes Average Particle Size Distribution of Standard Dust Across Different Injection Systems

## 5 Conclusions

In order to enhance the transmission efficiency of large particles within the Bio-SPAMS injection system while reducing its overall size, this study has developed a novel injection system, the PFW-ALens, designed to focus large particles. This injection system incorporates a new pre-focus structure, which effectively minimizes the dimensions of the buffer chamber and the number of lenses compared to traditional pre-focus injection systems, such as the work by Du et al. Furthermore, it maintains a low-loss transmission performance for a wide range of particle sizes. By integrating the pre-focus structure, the focusing capability of the five-stage lens system has been significantly improved to accommodate particles up to 10 μm.

The numerical simulation results indicate that the PFW-ALens is capable of focusing and transmitting particles within the size range of 100 nm to10 μm. Notably, the radial distribution of particles in the buffer chamber exhibits an inverse correlation with particle size after exiting the virtual impactor, a phenomenon not previously observed in earlier studies. The findings reveal that when particle sizes are less than 9 μm, the transmission efficiency can exceed 90 %. Particles within the range of 200 nm to 4 μm demonstrate a transmission efficiency of 100 %. The injection system designed in this study achieves the broadest particle size range and the highest transmission efficiency among systems with similar structural dimensions. This innovative design is conducive to further reducing the structural size of the injection system and the number of aerodynamic lenses, providing a foundation for the miniaturization of mass spectrometers.

*Data availability.* These data can be publicly accessible in free.

*Author contributions.* LL and ZXH designed the study; JHH and XL performed the

simulations and experiments; JHH, LL, XL, ZXH and ZC participated in data analysis
and result discussion; JHH and LL wrote the paper with the input from all authors.

*Competing interests.* The authors declare that they have no conflict of interest.

*Financial support.* This research was funded by the National key research and
development program for young scientists (2022YFF0705400) and the Fundamental
Research Funds for the Central Universities (21623205) .

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
