# Peer review of "Design of Wide Particle Size Range Aerodynamic Injection"

_EGUsphere, 2024_

## Referee Comment (RC1)

Referee report for "Design of Wide Particle Size Range Aerodynamic Inlet System with New Pre-focus Structure" by J. Huang et al, 2024

The study by Huang et al. proposes a new aerosol inlet design that is a miniaturized version of the Du et al., 2023 inlet, yet gives similar performance (though that is never clearly stated). The scientific contribution is significant, and the subject is appropriate for AMT. The organization is good (with the exception of Fig 3 and 6 being far separated), but the writing is often difficult to follow and contains extensive grammatical and repetition problems. Most often the meaning is conveyed, although some sentences are indecipherable.

Overall, this manuscript was submitted 1 or 2 editing versions too early. It contains two major discrepancies and extensive minor problems. Many labels are missing, figure captions are inadequate, critical details are excluded. A thorough effort in editing and re-writing is necessary to achieve the quality of its predecessor paper, Du et al. 2003.

That said, the contribution of the study (miniaturized aerosol inlet design with impressive performance) is significant enough that the authors be given the opportunity to resubmit the paper when the Major and Minor problems detailed below are addressed.

Major comments

1) There are multiple lines of evidence that put the experimental TE results of Fig 7, and the uncanny agreement with simulation, into question. A) Most of the values are exactly 100%, which is highly suspect, and include no error bars showing variability. B) As written, the reference system for submicron sizes (a CPC without a DMA) is incapable of accurately counting PSL generated by nebulization, assuming that many (typically *very* many) non-PSL (surfactant) particles are also generated. C) The simulated particle trajectories in Fig 6 show that particle beam widths at the detection laser(s) are much larger than a typical SPAMS laser beam width (Du et al., 2024). Therefore a large fraction of the simulated particles would not be detected by SPAMS, and experimental TEs cannot be at or near 100%. Furthermore, it is very likely that these simulated beam widths provide a lower limit, and the actual particle beam widths are probably much larger.

The authors need to re-analyze the Fig 7 results and/or convincingly justify the presented values by including a detailed data table, statistics, formulae, and a discussion of points A,B,C above.

2) There is much confusion surrounding the presentation of results for "new" and "old" systems. The new design for the current study is presented in Fig 1, the "PFW-Alens". The old system is that of Du et al., 2023. However, the simulations in Fig 5 present what is apparently a mix of the Du 2023 design (large buffer region, skimmer style VI) and the current design (5 focusing stages) for all five panels, despite that these are described in the text as the PFW-Alens. "New" and "old" designs are compared in Fig 5a & 5b, but these are again just the Du design with and without a pre-focuser. Confusingly, the final Du 2023 design contains a pre-focuser, so it's unclear what Fig 5b ("old") refers to. Consequently, the simulated transmission efficiency (TE) results of Fig 3 are also in question. Which design was simulated for the "Present Study" lines?

The authors need to make the results in Fig 3, 4, and 5 consistent. Simulation results for the present study/new design should all be based on the design in Fig 1.

Minor comments

General. The text needs another editing pass with a focus on sentence structure and verb tense. Examples of grammatical problems and repetition are lines 29-31, 89-92, 196-199, 232-233, 241-245, 303-305, although the problems occur throughout. Also, it is strange that simulated transmission efficiency is discussed in two separate places: section 3.1/Fig 3 and section 3.3/Fig 6. Consider combining those two Figures, e.g., as a single figure with two panels, along with their associated text. Also use consistent terminology throughout, e.g., choose either "inlet" or "injection system".

Line 18, 271, 309. Clearly this newly designed aerodynamic lens is not a "PM2.5" lens – it transmits much larger particles! Omit this terminology or explain.

Line 28. Define APS.

Line 52-60. The brief summary of previous aerosol inlets gives helpful context. However, the size ranges given seem arbitrarily chosen. For instance, while the Zelenyuk inlet transmits the given size range of 125-600nm at near 100% transmission efficiency (TE), their inlet transmits a much wider size range out to ~1.5um or larger at lower but still useful efficiency for SPMS studies. Modify the text to clarify the given range, for instance, if they denote the peak of the TE for each inlet, or the range where TE approaches 100%.

Line 104. It would be much clearer to the reader if the authors stated clearly that the purpose of the study is to design a miniaturized version of the Du et al., 2023 inlet that gives similar performance.

Line 125 Fig 1 shows a disc at the downstream end of the separation cone that provides an inner diameter reduction. What is the effect of that diameter reduction (presumably to 1.6mm – please add a label)? Does it provide a pressure drop so that separation cone acts as a better virtual impactor? Or is it a skimmer? Later in the manuscript (section 3.1) and also in the abstract, the authors refer to a virtual impactor, but it is not shown in Fig 1. Add a description of how the cone acts as a virtual impactor, and perhaps how it compares in design to Du 2023, which is quite different, and update your terminology in Section 2.1 accordingly.

Line 129. It seems important to mention that the accelerating/downstream nozzle has a tapered cone just upstream. Presumably, this reduces impaction, and it will also affect the trajectories of particles entering the particle sizing region. State any known effects of this converging nozzle design.

Section 2.1. It appears that Du et al., 2023 used a differentially pumped region between the aerodynamic lenses and the downstream nozzle, whereas the current study does not. State any ramifications of this design difference (is it just simpler?).

Line 139. I don't quite follow the pressure descriptions. Please clarify. Are the two pump-out lines on the "separation cone" fixed at 600 Pa? Is 300 Pa a fixed pressure, or is that the approximate pressure aerodynamic lens pressure that results from the calculation? Other parameters of interest to include would be the volume flow rate through the upstream critical orifice, the lens pressure, and the pressure drop down the lens stack.

Line 143.  Presumably the critical orifice region contains non-laminar and supersonic flow.  Does the viscous model treat this region appropriately?  Also state whether any symmetry was enforced for the simulations.

Lines 143-147.  Several sentences are poorly written.  Please re-write to clarify.  What is DPM?  What are cloud images?  What is the UDF that you refer to?

Section 2.3.  Mention why the additional airflow in the upper right of Fig 2 is needed and how it is used your TE calculation.

Line 154.  Give a model number or describe the ICPMS atomizer.

Line 167.  Define APS 3321.  Describe how aerodynamic diameter is measured.

Line 169-170.  The PMT detections for which laser, the first or second?  State what % of particles detected by the first PMT are also detected by the second PMT.

Line 172-173.  Presumably, you are also accounting for the flow rates into the instruments (and any dilution flow)?  State all the flow rates.

Line 175.  Nebulized PSL solutions typically produce polydisperse particles containing surfactant, which can greatly outnumber the PSL particles.  How are these non-PSL particles accounted for in the TE calculation, both for the supermicron and sub-micron setups, particularly since the CPC cannot distinguish between PSL and surfactant particles?  Can you show a size distribution of the generated particles?

Line 185.  It is not clear what the authors refer to as a "virtual impactor", particularly an inlet design that does not contain one.  For the "without VI" case, what part was removed in Fig 1?  Note that in general, *any* type of transverse pumpout design like the separation cone presented here will act as a VI to some extent for large particles.  Please clarify VI and without VI throughout this section.

Section 3.1.  Since virtual impactor is mentioned frequently, state the approximate particle size above which the VI becomes effective, ie, where nearly all particles are transmitted downstream and very few are being pumped out.  From Fig 3, it appears that ~90% of 100 nm particles are transmitted downstream and are *not* pumped out, correct?

Fig 3.  State that these are all (presumably?) simulations, not experimental results.  Add references for the without VI and without pre-focus lines.  How is TE defined here, eg, counting the particles exiting the nozzle, or those crossing the laser beams...?  It is difficult to see the data points that are on top of one another.

Line 237-239.  The argument about how particle acceleration in the pre-focus stages affects downstream divergence angle is not convincing, and I believe it is incorrect.  In both the Du and present designs, the particles achieve the same maximum velocities when passing through the critical hole, regardless of whether the pre-focus stages accelerate them or not.  Consider instead the difference in radial distributions between the Du pre-focus design and the current one (Fig 5a).  The strong downstream divergence in the Du design is due to the wider radial distribution and higher incident angle as particles

approaches the critical hole. Reconsider these arguments and update text accordingly, in this section and the abstract.

Line 239. Clarify "degree of change"

Fig 4. Axial velocity varies considerably across the diameter. Are these average velocities, centerline velocities, … ?
Also, add labels for the different parts of the inlet system at the appropriate axial ranges.

Fig 5. Can you comment on particle divergence in the vacuum region (downstream of the last nozzle), and how it changes with size? Presumably the converging nozzle design is advantageous here.
Also, add a velocity color scale, with consistent scaling for all panels.

Line 260. Why 110mm?

Line 265. State how the particle beam width compares with the beam waist (at focus) for both detection lasers.

Fig 6 & 7. Are the black lines the same as Fig 3 black line? Why are the simulated TE values different between Fig 6 and 7? The data are plotted versus "aerodynamic diameter", which is probably determined under the *vacuum* conditions of the SPAMS instrument, rather than *continuum* aerodynamic diameter as is typically defined. Clarify.

Line 291. Remove all-caps. List vendor. Also, mention something about the experimental sampling setup – is it identical to Fig 1 green lines.

Fig 8. Are the lines calculated as the average over seconds/minutes/hours? State in the caption. The APS line appears smoothed – is it? Clarify the y-axis – is it normalized dN/dlogD or normalized counts? Clarify "aerodynamic diameter", which I suspect is *continuum* for the APS and *vacuum* for SPAMS. If a conversion was done, what density and shape factor were applied? In general, it is reasonable to plot normalized size distributions as the authors have done here. However, it would be even more informative to plot absolute concentration, say as dN/dlogD, which then provides another demonstration of TE but now for non-spherical particles. Optional, at the authors' discretion.

Line 307. Add Du 2023 reference, just to be clear what is meant by "traditional".

References
Du, X.; Zhuo, Z.; Li, X.; Li, X.; Li, M.; Yang, J.; Zhou, Z.; Gao, W.; Huang, Z.; Li, L. Design and Simulation of Aerosol Inlet System for Particulate Matter with a Wide Size Range. Atmosphere 2023, 14, 664. https://doi.org/10.3390/atmos14040664

Du, X., Xie, Q., Huang, Q., Li, X., Yang, J., Hou, Z., Wang, J., Li, X., Zhou, Z., Huang, Z., Gao, W., and Li, L.: Development and characterization of a high-performance single-particle aerosol mass spectrometer (HP-SPAMS), Atmos. Meas. Tech., 17, 1037–1050, https://doi.org/10.5194/amt-17-1037-2024, 2024.

---

## Author Comment (AC1)

Answer for Major comments:

**Q1:** There are multiple lines of evidence that put the experimental TE results of Fig 7, and the uncanny agreement with simulation, into question. A) Most of the values are exactly 100%, which is highly suspect, and include no error bars showing variability. B) As written, the reference system for submicron sizes (a CPC without a DMA) is incapable of accurately counting PSL generated by nebulization, assuming that many (typically very many) non-PSL (surfactant) particles are also generated. C) The simulated particle trajectories in Fig 6 show that particle beam widths at the detection laser(s) are much larger than a typical SPAMS laser beam width (Du et al., 2024). Therefore a large fraction of the simulated particles would not be detected by SPAMS, and experimental TEs cannot be at or near 100%. Furthermore, it is very likely that these simulated beam widths provide a lower limit, and the actual particle beam widths are probably much larger.

**A1:** We thank the reviewer for the careful examination of the paper. For issues (a) and (b), the authors performed experimental tests on the transmission efficiency of PSL spheres using SMPS. The figure below shows the particle size distribution when generating 200 nm particle, indicating that the produced particles are primarily 200 nm, with no smaller particles generated. The calculation of transmission efficiency has been defined in Section 2.3, and specific test results are shown in Fig. 6, with error bars added based on the experimental results. Regarding issue (c), the authors would like to clarify that the Bio-SPAMS used in this study is not the same as Du et al.'s HP-SPAMS, and a different laser is employed. This study utilized the Sony SLD3234VF laser, which has a beam width of 3 mm, larger than the beam width shown in Fig. 6, thus it will not affect particle counting.

[Figure]

**Q2:** There is much confusion surrounding the presentation of results for "new" and "old" systems. The new design for the current study is presented in Fig 1, the "PFW-Alens". The old system is that of Du et al., 2023. However, the simulations in Fig 5 present what is apparently a mix of the Du 2023 design (large buffer region, skimmer style VI) and the current design (5 focusing stages) for all five panels, despite that these are described in the text as the PFW-Alens. "New" and "old" designs are compared in Fig 5a & 5b, but these are again just the Du design with and without a pre-focuser. Confusingly, the final Du 2023 design contains a pre-focuser, so it's unclear what Fig 5b ("old") refers to. Consequently, the simulated transmission efficiency (TE) results of Fig 3 are also in question. Which design was simulated for the "Present Study" lines?

**A2:** Thank you for the valuable comments. The systems have been renamed in the article. As introduced in Section 3.1, "this study first removed the virtual impactor and pre-focus structure from Fig. 1 and simulated the transmission efficiency of the model (represented by the blue left triangle line). Subsequently, the virtual impactor (orange diamond line, original design) and the pre-focus structure (black square line, present design) were sequentially reintroduced to observe the enhancements in transmission efficiency." Additionally, the model in Fig. 1 has been updated to include details such

as the virtual impactor to avoid any misunderstanding for the readers. Fig. 5 has also been modified accordingly, where Fig. 5(a) and (b) respectively show the transmission trajectories of 5 μm particles in both the design of this study and the original design, highlighting the advantages of the pre-focus structure in this study. Meanwhile, (c) and (d) display the transmission trajectories of 8 μm and 10 μm particles under the design of this study, demonstrating the advantages of this structure in transmitting larger particles. The line used in Fig. 3 (black square line) corresponds to the design shown in Fig. 1.

**Answer for Minor comments**

**Q1:** General. The text needs another editing pass with a focus on sentence structure and verb tense. Examples of grammatical problems and repetition are lines 29-31, 89-92, 196-199, 232-233, 241-245, 303-305, although the problems occur throughout. Also, it is strange that simulated transmission efficiency is discussed in two separate places: section 3.1/Fig 3 and section 3.3/Fig 6. Consider combining those two Figures, e.g., as a single figure with two panels, along with their associated text. Also use consistent terminology throughout, e.g., choose either "inlet" or "injection system".

**A1:** Thank you for identifying and correcting several spelling and grammatical errors in my article, which has improved its overall quality. The author has modified the sentence structures, verb tenses, and some complex sentences throughout the text, as well as standardized the term "injection system" for better readability. Regarding the transmission efficiency curves in the original Figures 3 and 6, the revised manuscript has combined these into a new Fig. 3.

**Q2:** Line 18, 271, 309. Clearly this newly designed aerodynamic lens is not a "PM2.5" lens – it transmits much larger particles! Omit this terminology or explain.

**A2:** We thank the reviewer for the careful examination of the paper. The revised manuscript has removed the description of PM2.5 lenses.

**Q3:** Line 28.    Define APS.

**A3:** We thank the reviewer for the careful examination of the paper. The revised manuscript has rewritten the description of APS as follows: "In the testing of standard dust, the wide-range particle size distribution obtained by the new injection system is highly consistent with Aerodynamic Particle Sizer (APS)".

**Q4:** Line 52-60. The brief summary of previous aerosol inlets gives helpful context. However, the size ranges given seem arbitrarily chosen. For instance, while the Zelenyuk inlet transmits the given size range of 125-600nm at near 100% transmission efficiency (TE), their inlet transmits a much wider size range out to ~1.5um or larger at lower but still useful efficiency for SPMS studies. Modify the text to clarify the given range, for instance, if they denote the peak of the TE for each inlet, or the range where TE approaches 100%.

**A4:** Thank you for your feedback. The transmission range mentioned in your comments is defined as the particle diameter range where the transmission efficiency exceeds 50%. The revised expression in the article is as follows: "We typically assess the particle transmission capacity of injection systems by considering the range where the transmission efficiency exceeds 50%.".

**Q5:** Line 104. It would be much clearer to the reader if the authors stated clearly that the purpose of the study is to design a miniaturized version of the Du et al., 2023 inlet that gives similar performance.

**A5:** Thank you for your feedback. In fact, although the injection system designed in this paper is similar in performance to that of Du et al., it outperforms their design in terms of the composition of the pre-focus structure and the performance of the particle beam width. Additionally, it employs a smaller buffer chamber and fewer stages of aerodynamic lenses. Therefore, it cannot be simply regarded as a miniaturized version of Du et al.'s design.

**Q6:** Line 125 Fig 1 shows a disc at the downstream end of the separation cone that

provides an inner diameter reduction. What is the effect of that diameter reduction (presumably to 1.6mm – please add a label)? Does it provide a pressure drop so that separation cone acts as a better virtual impactor? Or is it a skimmer? Later in the manuscript (section 3.1) and also in the abstract, the authors refer to a virtual impactor, but it is not shown in Fig 1. Add a description of how the cone acts as a virtual impactor, and perhaps how it compares in design to Du 2023, which is quite different, and update your terminology in Section 2.1 accordingly.

**A6:** Thank you for the valuable comments from the reviewer. The revised manuscript has adjusted Fig. 1, which now includes additional details such as the virtual impactor and the differential vacuum after the nozzle. The design of the virtual impactor is largely consistent with that of Du et al.

**Q7:** Line 129. It seems important to mention that the accelerating/downstream nozzle has a tapered cone just upstream. Presumably, this reduces impaction, and it will also affect the trajectories of particles entering the particle sizing region. State any known effects of this converging nozzle design.

**A7:** Thank you for the valuable comments from the reviewer. The revised manuscript has added a description of the conical nozzle and highlighted its advantages. The revised description is as follows: "In addition, this study utilized a smooth nozzle at the end of the aerodynamic lens. As mentioned by Zhang et al. (2004) in their study, this nozzle provides better collimation for small particles and improves the transport efficiency of large particles compared to stepped nozzles."

**Q8:** Section 2.1. It appears that Du et al., 2023 used a differentially pumped region between the aerodynamic lenses and the downstream nozzle, whereas the current study does not. State any ramifications of this design difference (is it just simpler?).

**A8:** Thank you for your feedback. In fact, this study also employs a differential pumping design between the nozzle and the vacuum chamber. To avoid any misunderstanding, the revised manuscript has added this detail in Fig. 1

**Q9:** Line 139. I don't quite follow the pressure descriptions. Please clarify. Are the two

pump-out lines on the "separation cone" fixed at 600 Pa? Is 300 Pa a fixed pressure, or is that the approximate pressure aerodynamic lens pressure that results from the calculation? Other parameters of interest to include would be the volume flow rate through the upstream critical orifice, the lens pressure, and the pressure drop down the lens stack.

**A9:** Thank you to the reviewer for your meticulous review; your suggestions have encouraged us to present our findings more effectively. In this study's simulation setup, the boundary pressure at the inlet is set to 101325 Pa, the boundary pressure at the pump outlet of the virtual impactor is set to 600 Pa, the boundary pressure at the nozzle outlet is set to 1 Pa, and the boundary pressure at the vacuum chamber outlet is set to 0.01 Pa. The mentioned pressure of below 300 Pa in the buffer chamber and lenses refers to the internal pressure under the constraints of the above boundary conditions. This explanation clarifies that the pressure conditions are intended to align with the study by Zhang et al. to ensure the accuracy of the model setup.

**Q10:** Line 143. Presumably the critical orifice region contains non-laminar and supersonic flow. Does the viscous model treat this region appropriately? Also state whether any symmetry was enforced for the simulations.

**A10:** Thank you to the reviewer for your meticulous review. In fact, the simulation conditions used in this study are consistent with those employed by Zhang et al. and Du et al. in their numerical simulations, both utilizing a laminar flow model for calculations. Additionally, to reduce computation time, the model used in this study is a two-dimensional axisymmetric model.

**Q11:** Lines 143-147. Several sentences are poorly written. Please re-write to clarify. What is DPM? What are cloud images? What is the UDF that you refer to?

**A11:** We are grateful for the reviewer's recommendations; your input has clarified several key points in our research. DPM refers to the Discrete Phase Model, which is used to simulate the motion of discrete phases (such as particles) in a fluid. The term "cloud images" was incorrectly translated and has been changed to "contours." UDF

stands for User-Defined Functions, which add volume forces such as Brownian and drag forces to Fluent through a compiled program, allowing for more accurate particle motion.

**Q12:** Section 2.3. Mention why the additional airflow in the upper right of Fig 2 is needed and how it is used your TE calculation.

**A12:** Thank you to the reviewer for your valuable insights; we have incorporated your suggestions to enhance the overall quality of our paper. In Fig. 2, the airflow in the upper right corner is primarily intended to match the flow rate differences among the experimental devices. As mentioned in the text: "It is important to note that during the experiment, we utilized additional airflow. For experiments involving particles smaller than 1 μm, this additional airflow was implemented to..." By balancing the flow rates of the various devices, this study calculated the transmission efficiency using the ratio of the number of particles.

**Q13:** Line 154. Give a model number or describe the ICPMS atomizer.

**A13:** Thank you for your feedback. The model of the atomizer has been added, and the revised manuscript describes it as follows: "Initially, the PSL solution is diluted with pure water, after which nitrogen serves as the carrier gas to atomize the PSL solution using an ICPMS atomizer (Ge, C21-1-UFT02)."

**Q14:** Line 167. Define APS 3321. Describe how aerodynamic diameter is measured.

**A14:** Thank you to the reviewer for your expert feedback; your recommendations have been beneficial in refining our conclusions. The revised manuscript has been changed to APS3321. Additionally, the diameter measurement method used by Bio-SPAMS is similar to that of APS, so the principles of laser diameter measurement for both have also been included in the text. Specifically, it states: "The first diameter measuring laser is designed with a beam-splitting optical structure similar to that of the APS 3321. The method involves using a beam splitter to divide the diameter measuring laser (Sony SLD3234VF) into two nearly parallel beams and calculating the aerodynamic diameter

of the particles by the time they pass through these beams."

**Q15:** Line 169-170. The PMT detections for which laser, the first or second? State what % of particles detected by the first PMT are also detected by the second PMT.

**A15:** Thank you for your feedback. PMT1-1 is used to detect the number of particles. Therefore, the calculation of transmission efficiency in this study is based on the counts from PMT1-1. During the research process, we found that there was essentially no loss in transmission between the two PMTs, and this result can also be validated by the transmission trajectory shown in Fig. 5.

**Q16:** Line 172-173. Presumably, you are also accounting for the flow rates into the instruments (and any dilution flow)? State all the flow rates.

**A16:** Thank you for your feedback. The revised manuscript has labeled the flow rate into the instrument in Fig. 2.

**Q17.** Line 175. Nebulized PSL solutions typically produce polydisperse particles containing surfactant, which can greatly outnumber the PSL particles. How are these non-PSL particles accounted for in the TE calculation, both for the supermicron and sub-micron setups, particularly since the CPC cannot distinguish between PSL and surfactant particles? Can you show a size distribution of the generated particles?

**A17:** Thank you to the reviewer for your meticulous review. This study utilized SMPS to test the transmission efficiency of small particles. The response to question 1 also confirmed that no other particles were produced to affect the count when measuring the 200 nm particle. Other experimental test results are shown in Fig. 6, where the lower transmission efficiency for the 100 nm particles is due to the Bio-SPAMS used in this study having fluorescence detection capability. As a result, the energy of the two beams of the first laser was adjusted to a ratio of 70:30, which led to a weaker bimodal signal for the 100 nm particles, ultimately resulting in a lower efficiency.

**Q18:** Line 185. It is not clear what the authors refer to as a "virtual impactor", particularly an inlet design that does not contain one. For the "without VI" case, what

part was removed in Fig 1? Note that in general, any type of transverse pumpout design like the separation cone presented here will act as a VI to some extent for large particles. Please clarify VI and without VI throughout this section.

**A18:** Thank you to the reviewer for your meticulous review. The revised manuscript has added a schematic diagram of the virtual impactor structure in Fig. 1.

**Q19:** Section 3.1. Since virtual impactor is mentioned frequently, state the approximate particle size above which the VI becomes effective, ie, where nearly all particles are transmitted downstream and very few are being pumped out. From Fig 3, it appears that ~90% of 100 nm particles are transmitted downstream and are not pumped out, correct?

**A19:** We deeply appreciate the reviewer's constructive criticism; your insights have helped us address potential oversights in our work. Simulation results indicate that the VI achieves a transmission efficiency of nearly 90 % for particles larger than 100 nm. Additionally, the comparison without VI and the pre-focusing sampling system is shown in Fig. 3, depicted by the blue left triangle curve. The definition of transmission efficiency has been added to the text as follows: "The transmission efficiency presented in this study is the ratio of the number of particles at a distance of 110 mm from the lens outlet to the number of particles at the inlet of the sampling system."

**Q20:** It is difficult to see the data points that are on top of one another.

**A20:** Thank you to the reviewer for your meticulous review; your suggestions have encouraged us to present our findings more effectively. To increase the visibility of overlapping data points, the symbols of some curves are changed to hollow.

**Q21:** Line 237-239. The argument about how particle acceleration in the pre-focus stages affects downstream divergence angle is not convincing, and I believe it is incorrect. In both the Du and present designs, the particles achieve the same maximum velocities when passing through the critical hole, regardless of whether the pre-focus stages accelerate them or not. Consider instead the difference in radial distributions between the Du pre-focus design and the current one (Fig 5a). The strong downstream

divergence in the Du design is due to the wider radial distribution and higher incident angle as particles approaches the critical hole. Reconsider these arguments and update text accordingly, in this section and the abstract.

**A21:** Appreciation is extended for the reviewer's valuable comments. Regarding the distribution of axial velocity, it is indeed true that, regardless of the structure of the injection system, particles achieve the same maximum velocity when passing through the critical orifice. To facilitate a better comparison of the performance of different systems, the comments have been taken into account, and the velocity distribution plot has been changed to a comparison of the beam widths of different particles at various positions within the sampling system, along with an analysis, as shown in Fig. 4.

**Q22:** Line 239. Clarify "degree of change"

**A22:** Thank you to the reviewer for your meticulous review. The revised manuscript has removed the description of the impact of the pre-focusing structure on particle velocity. In the previous version, this section aimed to demonstrate that the addition of a pre-focus structure could improve the transmission range of the lens from 5 μm to 10 μm.

**Q23:** Fig 4. Axial velocity varies considerably across the diameter. Are these average velocities, centerline velocities, … ? Also, add labels for the different parts of the inlet system at the appropriate axial ranges.

**A23:** We appreciate the reviewer's thorough evaluation; your constructive remarks have guided our revisions significantly. The velocity distribution map has been deleted, and the revised manuscript has added a comparison of beam widths for different particles at different positions in the injection system, as shown in Fig. 4.

**Q24:** Fig 5. Can you comment on particle divergence in the vacuum region (downstream of the last nozzle), and how it changes with size? Presumably the converging nozzle design is advantageous here.

**A24:** We greatly appreciate the reviewer's comments; your guidance has been

invaluable in shaping the final version of our manuscript. The injection system in this study, under the influence of the newly designed pre-focusing structure, significantly reduces the particle beam width. After focusing through five stages of lenses, the particles exhibit minimal noticeable divergence upon exiting the converging nozzle, as demonstrated by the full movement trajectories at 8 μm (Fig. 5(e)) and 10 μm (Fig. 5(f)). In fact, recent simulation studies have found that reducing the number of lens stages to three can still achieve the transmission range presented in this study.

**Q25:** Also, add a velocity color scale, with consistent scaling for all panels.

**A25:** Thank you to the reviewer for your expert feedback. A colormap has been added; however, Fig. 5 primarily aims to illustrate the loss of particle transmission, for which the particle ID colormap has been selected. The color scale allows for understanding where the losses occur for the particles. As for the differing scale ratios in (e) and (f), this is intended to showcase the global flow trajectories of larger particles.

**Q26:** Line 260. Why 110mm?

**A26:** Thank you to the reviewer for your meticulous review; your suggestions have encouraged us to present our findings more effectively. In fact, all lasers are located at a position less than 110 mm from the nozzle outlet, so the author chose 110 mm to demonstrate that good beam width and transmission efficiency can still be maintained at a distance from the laser.

**Q27:** Line 265. State how the particle beam width compares with the beam waist (at focus) for both detection lasers.

**A27:** Thank you to the reviewer for your expert feedback. By analyzing Fig. 5 and Q26, it can be determined that there is no noticeable divergence or expansion of the particle beam after exiting the nozzle.

**Q28:** Fig 6 & 7. Are the black lines the same as Fig 3 black line? Why are the simulated TE values different between Fig 6 and 7? The data are plotted versus "aerodynamic

diameter", which is probably determined under the vacuum conditions of the SPAMS instrument, rather than continuum aerodynamic diameter as is typically defined. Clarify.

**A28:** We appreciate the reviewer's thorough evaluation; your constructive remarks have guided our revisions significantly. The differences in TE between the simulations may be attributed to the author not selecting the same number of particle beams while processing data on different computers. The author has standardized the displayed number of particles and modified the values in the three figures. As for the X-axis in the figures, the revised manuscript has also adjusted it to represent particle diameter. However, it should be noted that the Bio-SPAMS used in this study also employed particle size calibration in the particle size measurement section, determining the aerodynamic diameter of the particles based on their flight time.

**Q29:** Line 291. Remove all-caps. List vendor. Also, mention something about the experimental sampling setup – is it identical to Fig 1 green lines.

**A29:** Thank you to the reviewer for the suggestion; the supplier of the standard dust sample has been added. The revised text is as follows: "To characterize the analytical capabilities of the PFW-ALens for large particles, standard ultrafine dust (ISO 12103, PTI) was selected as the test sample." The experimental sampling setup is consistent with the green line in Fig. 2.

**Q30:** Fig 8. Are the lines calculated as the average over seconds/minutes/hours? State in the caption. The APS line appears smoothed – is it? Clarify the y-axis – is it normalized dN/dlogD or normalized counts? Clarify "aerodynamic diameter", which I suspect is continuum for the APS and vacuum for SPAMS. If a conversion was done, what density and shape factor were applied? In general, it is reasonable to plot normalized size distributions as the authors have done here. However, it would be even more informative to plot absolute concentration, say as dN/dlogD, which then provides another demonstration of TE but now for non-spherical particles. Optional, at the authors' discretion.

**A30:** Thank you to the reviewer for your expert feedback; your recommendations have

been beneficial in refining our conclusions. These lines are calculated based on a 3-minute average and have been added to the title, which now reads: "3-Minutes Average Particle Size Distribution of Standard Dust Across Different Injection Systems." Additionally, the Y-axis represents dN/dlogD, while the X-axis is selected as particle diameter, which has been calibrated based on flight time.

**Q31:** Line 307. Add Du 2023 reference, just to be clear what is meant by "traditional"

**A31:** We greatly appreciate the reviewer's comments. The mention of the design by Du et al. has been added in the conclusion to improve readability. The revised text is as follows: "This injection system incorporates a new pre-focus structure, which effectively minimizes the dimensions of the buffer chamber and the number of lenses compared to traditional pre-focus injection systems, such as the work by Du et al.". Regarding the two references mentioned by the reviewer, namely Du, X.; Zhuo, Z.; Li, X.; Li, X.; Li, M.; Yang, J.; Zhou, Z.; Gao, W.; Huang, Z.; Li, L.'s "Design and Simulation of Aerosol Inlet System for Particulate Matter with a Wide Size Range. Atmosphere 2023, 14, 664. https://doi.org/10.3390/atmos14040664" and Du, X., Xie, Q., Huang, Q., Li, X., Yang, J., Hou, Z., Wang, J., Li, X., Zhou, Z., Huang, Z., Gao, W., and Li, L.'s "Development and characterization of a high-performance single-particle aerosol mass spectrometer (HP SPAMS), Atmos. Meas. Tech., 17, 1037–1050, https://doi.org/10.5194/amt-17-1037-2024, 2024," they have been referenced in the previous version of the manuscript.

---

## Author Comment (AC2)

**Q1:** The introduction would benefit from a brief description of what the parts of the inlet system/aerodynamic lens are and what each one does. such as the critical orifice, the buffer region, the apertures in the aerodynamic lens, etc. It would also help if the nomenclature is consistent between the figures and the text and between different parts of the text. Each part that is called out in the text should be labelled in Figure 1. For example, a virtual impactor is mentioned in line 139 but I can't tell if that is the same thing as the pre-focus hole or the buffer chamber in Figure 1 or something else entirely. Where is the critical orifice and what diameter is it?

**A1:** Thank you to the reviewer for your helpful suggestion, it led to meaningful improvements in our manuscript. Labels for each section, such as the aerodynamic lens, have been added to Figure 1, and the naming of each part has been checked in the text for consistency. For example, the injection system is now uniformly referred to as "injection system." In Section 2.1 of the manuscript, the roles of the various components in the injection system are described, along with the critical orifice diameter of 0.26 mm. Additionally, the structure of the virtual impactor has been added to Fig. 1, which is the same as that used by Du et al.

**Q2:** What is the pressure in each stage of the inlet? What does "low-pressure loss" mean in lines 106-107?

**A2:** We appreciate the reviewer's careful reading and constructive suggestions; they have positively influenced our manuscript's development. The pressure settings have been introduced in Section 2.2. In the simulation setup of this study, the boundary pressure at the inlet is set to 101325 Pa, the boundary pressure at the pumping outlet of the virtual impactor is set to 600 Pa, the boundary pressure at the nozzle outlet is set to 1 Pa, and the boundary pressure at the vacuum chamber outlet is set to 0.01 Pa. The pressure referred to for the buffer chamber and lens being below 300 Pa indicates the pressure inside the buffer chamber and lens under the aforementioned boundary conditions. This pressure condition is specified to align with Zhang et al.'s study to ensure the accuracy of the model setup. The description of low-pressure loss was originally intended to indicate that no acceleration of particles occurs as they pass

through the pre-focusing structure, but to avoid any misunderstanding, the low-pressure loss has now been removed in the revised manuscript.

**Q3:** The authors mention multiple times that their goal is miniaturization of the SPAMS. Making the inlet smaller is not going to accomplish that. The size is really determined by the pumps and the mass spectrometers. It is ok to say that the goal is to extend the range of sizes transmitted by the inlet system.

**A3:** Thank you to the reviewer for the insightful suggestions. In fact, after utilizing the novel pre-focusing structure designed in this study, the width of the particle beam can be significantly reduced, allowing for the use of smaller buffer chambers and fewer stages of lenses. This reduction decreases the volume of the injection system. Since the radial length of SPAMS primarily depends on the volume of the injection system, I believe this will help in reducing the overall volume of the SPAMS instrument.

**Q4:** The abstract is confusing. There are multiple size ranges and it is not clear how they relate to each other. There are yet more size ranges in lines 110-113.

**A4:** Thank you to the reviewer for your constructive comments; we have carefully considered your recommendations in our revisions. The description of transmission efficiency has been revised, with redundant explanations removed. The updated text is as follows: "The numerical simulation results demonstrate that the particle transmission efficiency is greater than 90% for sizes ranging from 100 nm to 9 μm."

**Q5:** I do not understand how you can count PSL particles with the CPC without also counting the surfactant particles from the atomized solution.

**A5:** We sincerely appreciate the reviewer's thoughtful feedback. I would like to thank the reviewer for the careful examination of this paper. The authors conducted experimental tests on the transmission efficiency of PSL spheres using SMPS, and error bars have been added based on the experimental results. The figure below shows the particle size distribution when generating 200 nm particles, indicating that the produced particles are predominantly 200 nm in size, with no other smaller particles generated.

[Figure]

**Q6:** Lines166-170: What wavelength is the laser and what is the smallest size particle that can be detected?

**A6:** Thank you to the reviewer for your expert feedback. This study uses a 405 nm semiconductor laser with a wavelength of 405 nm. In the dual-beam design of Bio-SPAMS, the minimum detectable size is 100 nm.

**Q7:** Lines 183-185. Are the authors saying that all of those references added a buffer chamber and a virtual impactor? That is not true. Please read those papers and cite them correctly.

**A7:** Thank you to the reviewer for your keen insights. This citation is simply to demonstrate the wide range of applications for the lens designed by Zhang et al. In this study, the design enhances the transmission range to 10 μm by adding a virtual impactor, without requiring any changes to the lens dimensions proposed by Zhang et al., thereby providing greater possibilities for research across various fields.

**Q8:** Figure 3. The legend is confusing. Put the year in for Du et al. Are the bottom three all from the present study with different configurations? Or from previous studies? It is not clear.

**A8:** Thank you to the reviewer for your constructive comments; we have carefully considered your recommendations in our revisions. First, Fig. 3 illustrates the advantages of different structures (such as the virtual impactor and the pre-focus structure) through pairwise comparisons. Additionally, the explanations regarding the different systems in Fig. 3 have been provided in Section 3.1. As stated in the text: "Subsequently, the virtual impactor (orange diamond line, original design) and the pre-focus structure (black square line, current design) were sequentially reintroduced to observe the enhancements in transmission efficiency. By comparing the transmission effects of the three designs above, the advantages of the design in Fig. 1 are highlighted."

**Q9:** Line 212: What is a "strong" buffer?

**A9:** Thank you to the reviewer for your expert feedback. The original comparison of speeds in Fig. 3 has been replaced with a comparison of the widths of particle streams at different positions in the various sampling systems. Therefore, the original description has been revised and no longer exists.

**Q10:** Figure 4. What does the axial velocity impact in terms of particle transmission? What part of the diagram in Figure 1 does the grey box correspond to? Does it really make a difference that the blue line has a few wiggles in the grey box? The transmission efficiency shown in Figure 3 is the same for Du et al. and present study for those sizes. Do the authors have data for 9 or 10 micron particles? That would be more relevant since there is a difference in transmission efficiency.

**A10:** Thank you to the reviewer for your helpful suggestions; they led to meaningful improvements in our manuscript. The argument regarding how particle acceleration in the pre-focus structure affects the downstream divergence angle is incorrect. In other designs (such as the design by Du et al.), the noticeable divergence is due to the wider radial distribution and higher incidence angles of particles as they approach the critical

orifice. Therefore, the comparison of speeds has been changed to a comparison of beam widths. Additionally, global flow trajectory comparisons for 8 μm (e) and 10 μm (f) have been added to Fig. 5.

**Q11:** Figure 5. Why show both 5 and 6 micron particles? They are very similar. Panels a) and c) are both the "new" system but have different structures in the inset. Why? It's not clear what the authors are referring to with "new" and "old." Is "old" Du et al.? How does the structure in panel e} relate to Figure 1? They look different.

**A11:** We appreciate the reviewer's careful reading and constructive suggestions; they have positively influenced our manuscript's development. Fig. 5 is designed to show the distribution of particles larger than 5 μm before and after the virtual impactor, primarily to illustrate that there is no significant increase in beam width as the particle diameter increases. Additionally, the different structure of the illustrations is due to screenshot issues and has been modified. The "old" refers to the design by Du et al., and the revised manuscript has provided clear descriptions for the different systems. Fig. 1 is a schematic representation of particle transmission effects, and some details (such as the conical changing diameter interface before and after the buffer chamber) were not fully displayed; the revised manuscript has supplemented details such as the virtual impactor and changing diameter interface.

**Q12:** Figure 6. I would not label every point with its value. I think it makes the figure too busy. It would be better to put the data values in a table. Same comment for Figure 7. Scale the y-axis in Figure 7 from 0.

**A12:** We are thankful to the reviewer for the valuable input; your suggestions have been instrumental in refining our work. The Y-axis of Fig. 7 has been modified, and the original Fig. 6 has been merged with Fig. 3 and the values in the figure have been removed.

**Q13:** Line 260: Why place the target 11 cm downstream?

**A13:** Thank you to the reviewer for your keen insights. In fact, all lasers are located at

a position less than 110 mm from the nozzle outlet, so the author chose 110 mm to demonstrate that good beam width and transmission efficiency can still be maintained at a distance from the laser.

**Q14:** Lines 286-287. The authors do not know this. It is very common for experimental measurements of transmission efficiency to be different from calculations.

**A14:** Thank you to the reviewer for your expert feedback. The relevant description has been removed from the revised manuscript.

**Q15:** Line 309: What do the authors mean by "the 2.5 micron lens system?" Is this the same thing as the "aerodynamic five stage lens group" on line 121 and the "PM2.5 five stage lens" on line 271? Please use consistent names throughout the paper.

**A15:** We appreciate the reviewer's careful reading and constructive suggestions; they have positively influenced our manuscript's development. The description of the 2.5 μm lens was incorrect. To avoid misinterpretation, all instances of "2.5 μm lens" in the text have been changed to "five-stage lens."

**The answer for minor comments:**

**Q16:** Line 287: What is the B-SPAMS? Is it the same instrument mentioned previously? Then use the same acronym. Same comment for line303.

**A16:** We are thankful to the reviewer for the valuable input; your suggestions have been instrumental in refining our work. Bio-SPAMS refers to the biological aerosol single particle mass spectrometer, which is the single particle mass spectrometer used in this study and is not consistent with Du et al.'s HP-SPAMS. Additionally, the descriptions in the text have been standardized.

**Q17:** Line 146: Define DPM

**A17:** Thank you to the reviewer for your expert feedback. DPM refers to the Discrete Phase Model, which is used to simulate the motion of discrete phases (such as particles) in a fluid.

**Q18:** Line 291: Do not capitalize ULTRAFINE TEST DUST. Define APS. Give manufacturer and model.

**A18:** Thank you to the reviewer for the suggestion. The supplier of the standard dust sample has been added. The revised text is as follows: "To characterize the analytical capabilities of the PFW-ALens for large particles, standard ultrafine dust (ISO 12103, PTI) was selected as the test sample."

---

## Author Response (AR2)

**Answer for reviewers' comments**

Respond for RC1:

**Q1:** It is awkward that the single-particle MS used in this study ("Bio-SPAMS") is a new instrument but is not fully described here, nor referenced. "Bio-SPAMS used in this study is not the same as Du et al.'s HP-SPAMS, and a different laser is employed." Can you add to that basic description? Presumably this instrument is designed to detect (large) biological particles?

**A1:** We thank the reviewer for the careful examination of the paper. To enhance the readability of the article, the author added a description in Section 2.3 regarding the optical design of Bio-SPAMS, highlighting the differences between the Bio-SPAMS used in this study and HP-SPAMS, particularly its biofluorescence detection capability. The details are as follows "Additionally, Bio-SPAMS introduces a laser-induced biofluorescence detection module, enabling the pre-screening of fluorescent particles. This allows for selective ionization of only the fluorescent particles, meeting the requirements for online monitoring of bioaerosols."

Due to Bio spams being a newly designed instrument, relevant articles are still being written

**Q2:** Fig 1. Add units to caption. Also, I think most small TSI OPC models use a fixed 2.8 lpm flow rate, ie, much higher than Bio-SPAMS – please confirm.

**A2:** Thank you for the valuable comments. The flow rate annotations in Fig. 2 have been revised, and the unit (L/min) has been added to the figure caption.

**Q3:** Fig 1/Section 2.1. Define "virtual impactor" here, as it is not currently listed in this section. It is necessary to clarify the authors' terminology because all injection systems employing the pressure-reduction orifice plus transverse pumpout design (common in many/all the other aerosol MS systems referenced in the paper) inherently include a virtual impactor, ie, the sample air is enhanced with larger particles compared to the pumpout air. I believe the authors mean that the addition of the "separation cone" is

what defines a "virtual impactor" in this study, although technically the injection system already includes virtual impaction effect without the separation cone.

**A3:** Thank you for your feedback. To avoid confusion, the author revised "separation cone" to "virtual impactor" in the manuscript. Additionally, to enhance the readability of the article, the following statement was added: "In this study, the virtual impactor is defined as a device that enhances the concentration of larger particles in the sample air by employing a pressure-reduction orifice and transverse pumpout design. The addition of a separation cone further refines the virtual impaction effect, distinguishing it from the inherent virtual impaction observed in other aerosol MS systems referenced in the literature."

**Q4:** Also, list approx. pressures of all inlet components.

**A4:** Thank you to the reviewer for your meticulous review. In the revised manuscript, the author has added pressure annotations to Fig. 1.

**Q5:** Also, briefly describe the 2nd pumping stage downstream of the nozzle. Are one or both of the pumping lines fixed at a constant pressure, eg, using a commercial pressure controller?

**A5:** Thank you for your valuable feedback. In the revised manuscript, the author has added the pressure control method downstream of the nozzle, as described below: "Downstream of the nozzle, a second pumping stage is employed to further reduce the pressure and enhance particle acceleration into the vacuum chamber. This stage utilizes a molecular pump to achieve the necessary pressure drop, ensuring efficient particle transport and stable operation. Both pumping lines are maintained at a constant pressure using commercial pressure controllers, which provide precise control of the flow dynamics."

**Q6:** Line 120. A pressure reduction (critical) orifice is missing from this list. Also add labels in Fig 1 for separation cone and acceleration nozzle.

**A6:** Thank you for your valuable feedback. In the revised manuscript, the author has

added descriptions of the critical hole in the key components section, modified the annotations in Fig. 1 to include labels for the acceleration nozzle, and replaced all references to the separation cone with the term "virtual impactor".

**Q7:** Line 132. "Smooth" = tapered? Please clarify.

**A7:** To avoid confusion, the author has revised the description from "smooth nozzle" to "tapered nozzle."

**Q8:** Section 2.2. Please state whether the model considers compressible flow (which is well known to occur downstream of each critical orifice and, importantly, through which all particles pass). State in the main text what symmetry was employed.

**A8:** Thank you to the reviewer for your meticulous review. To enhance the rigor of the article, the author has added the following description: "The numerical simulation employs an ideal gas as the material property, which means it considers the characteristics of compressible flow, particularly as the gas undergoes rapid expansion and acceleration downstream of the critical hole. In this region, the compressible Navier-Stokes equations are used to resolve pressure gradients, density variations, and inertia effects, all of which are essential for capturing transitions between supersonic and subsonic flow, as well as shock phenomena. To reduce computational complexity, axial symmetry is rigorously applied, assuming that flow properties and particle trajectories are symmetric around the central axis of the orifice.

**Q9:** Line 142. Do the authors mean that the pressures are chosen based on numerical simulations of this injection system from Zhang's code? Please clarify.

**A9:** Thank you to the reviewer for your meticulous review. The selection of the physical model is indeed consistent with the design of Zhang et al., but the pressure values were chosen based on the design requirements of our instrument. To avoid misinterpretation, the author has moved this statement to a later section, where it is used to describe the related physical equations.

**Q10:** Line 162. A "three-way flow splitter"?

**A10:** We are grateful for the reviewer's recommendations. In the revised manuscript, the author has updated the relevant descriptions.

**Q11:** Line 178. APS = Aerodynamic Particle Sizer (TSI, Inc).

**A11:** We are grateful for the reviewer's recommendations. In the revised manuscript, the author has updated the relevant descriptions.

**Q12:** Line 179. For this new optical system, do the particle detection beams pass through focusing or shaping optics? State the approx. laser beam waists where the particles intersect the laser beams.

**A12:** In the novel optical system, the laser beam is shaped by optical components to form a rectangular spot of 600 μm × 30 μm. In this experiment, the particle beam width was measured at a position 110 mm downstream from the lens exit, which exceeds the distance between the lens and the beam profiling laser (67 mm), indicating that the actual particle beam width at the beam profiling laser position is narrower. Analysis of the microscope-captured distribution image of mixed particles (300 nm, 740 nm, 1 μm, 2 μm, 5 μm) at the beam profiling laser position (as shown in the figure below) reveals that the particle beam width at this location is smaller than the laser beam waist.

[Figure]

**Q13:** Line 185. Does the TE calculation also consider the flow difference?

**A13:** Thank you to the reviewer for your expert feedback. During the experimental pipeline connection, flow rate discrepancies between instruments were carefully considered, and a flow-balancing design was implemented to ensure consistent flow rates across all nodes.

**Q14:** Line 206. Again related to the "virtual impactor", please clarify what is actually being removed for this comparison (I think it is just the separation cone and not, eg, the entire upstream pumpout region).

**A14:** Thank you to the reviewer for your expert feedback. To prevent terminological ambiguity, the authors have revised the description of the "separation cone" to "virtual impactor" in the revised manuscript. Additionally, the statement regarding "removal of the virtual impactor and pre-focus structure" specifically indicates that within the flow path from the inlet to the buffer chamber, only the critical hole remain as core components, with all auxiliary structures being eliminated.

**Q15:** Fig 3. It surprising that the addition of a separation cone (aka "virtual impactor") has no theoretical effect on small particles (blue vs orange lines). One would expect that by adding the separation cone to enhance the inherent virtual impaction effect that is already there, the impactor's cutpoint diameter would shift to smaller sizes. But the blue line is already at 100% for the smallest particles shown. So although line 210 is technically correct ("Our team discovered that the virtual impactor used in this study is capable of transporting 100 nm particles downstream with an efficiency of over 90 %"), the same is apparently true for the injection system without the "virtual impactor". Consider clarifying.

**A15:** We deeply appreciate the reviewer's constructive criticism. First, the authors have standardized the terminology by revising the description of the "separation cone" to the standardized term "virtual impactor". Furthermore, in Section Q3, the functional role of this virtual impactor has been clarified: its primary purpose is inertial enrichment of large particles, whereas the 100 nm particle size threshold significantly exceeds the

effective cutoff for size-selective classification by virtual impaction. In other words, the operational objective of the virtual impactor in this design is not to expel small particles via pumping mechanisms, but rather to selectively concentrate large particles to optimize aerosol transmission efficiency within the sampling system.

**Q16:** Para starting line 217. Similarly, given the above interpretation of the enhanced virtual impaction effect, explain why the "virtual impactor has increased its ability to focus on large particles". If the VI cutpoint diameter was indeed reduced as I suspect, one would expect the large particle transmission to be relatively unchanged. The simulated D50 cutpoints for the two compared designs (with and without separation cone) are easy to estimate and may help illustrate.

**A16:** Thank you to the reviewer for your meticulous review. As detailed in the technical responses to Q15 and Q3, the virtual impactor in this sampling system not only achieves inertial enrichment efficacy for large particles through concentration enhancement, but also improves the flow-field distribution characteristics of large particles by optimizing aerosol dynamics, thereby effectively boosting the overall transmission efficiency. The newly added explanation states: "This improvement is primarily attributed to the virtual impactor, which refines the flow dynamics, reduces particle loss, and optimizes the impaction process, thereby enhancing the focusing efficiency for larger particles."

**Q17:** Lines 266-274. Most of this is repetitive with the previous paragraphs.

**A17:** We appreciate the reviewer's thorough evaluation. In the revised manuscript, the authors have implemented systematic optimizations. The newly states: "As shown in Fig. 4, the PFW-ALens-equipped injection system significantly reduces the radial width of the particle beam at various positions across different particle diameters, outperforming both the original design and Du et al.'s pre-focusing structure. Specifically, at the buffer chamber, the beam width is reduced by 70 % to 95 % compared to Du et al.'s design. Additionally, the radial distribution of particles in the buffer chamber exhibits an inverse correlation with particle size, a novel observation not seen in previous pre-focusing designs. This discovery offers a strong basis for

optimizing the lens system's length and minimizing the number of lenses required."

**Q18:** Fig 5. The colors are inverted. Define the color bar and add units.

**A18:** Regarding the colorbar in Fig. 5, it actually represents the particle ID numbers, which are systematically distributed according to the particles' initial radial positions. Therefore, the numerical orientation of the colorbar is correct and consistent with the design logic. To enhance clarity, we have added an explicit definition of the colorbar in the figure.

**Q19:** Line 293. Enhancement compared to what?

**A**19: We greatly appreciate the reviewer's comments. The author has updated the relevant statements in the revised manuscript as follows: "Fig. 3(b) demonstrates a remarkable performance in the particle transmission range of the PFW-ALens."

**Q20:** Line 297-299. This is incorrect. The Du et al. 2023 system has a beam waist of 300 microns at the particle beam. Were the Du optical design used on the current system, it appears that a large fraction of the particles (having radial width ~0.5-0.6 mm, Fig 3b) would not intersect the laser beams. No details are given regarding the "improved" optical design employed here (line 179), though it is likely that new scattering laser beams are focused with converging lenses, as is done in other single-particle MS systems, such that the beam waists where particles pass through them may or may not be smaller than the simulated particle beam width. Although technically this consideration does not affect the transmission efficiencies reported here (but is highly relevant to experimental detection efficiencies in Fig 7), some context regarding the laser beam versus particle beam widths for the current system is necessary. Correct and reword as it pertains to the current system.

**A20:** We greatly appreciate the reviewer's comments. Although the optical system design in this study aligns with that of Du et al. (2021), we have implemented critical optimizations by incorporating focusing element and beam-shaping components to precisely modulate the output beam into a 600 μm × 30 μm rectangular spot. Regarding

the observed variations in transmission efficiency in Figure 6, we have supplemented the revised manuscript with detailed analysis and experimental validation, further elucidating the optimization effects of beam shaping on aerosol transmission efficiency. Specifically, as follows "Although a beam width of 600 μm was employed in this study, the particle beam width for certain small particles still exceeds this threshold. According to Rayleigh scattering theory, the scattered light intensity is proportional to the sixth power of the particle diameter, leading to a dramatic decline in signal intensity as the particle size decreases. Due to the combined effects of these two factors, the detection sensitivity and transmission efficiency of small particles (e.g., 100 nm) are significantly compromised, as indicated by the low transmission efficiency of 100 nm particles in Fig. 6."

**Q21:** Fig 6. The red line is actually detection efficiency, which in addition to particle transmission also includes the efficiency of particle detection by light scattering. Also, add a caption and describe error bars.

**A21:** Thank you to the reviewer for your meticulous review. The error bars are plotted based on statistical data from five independent experiments, with the upper and lower bounds representing the maximum and minimum measured values, respectively, and the midpoint denoting the average value across these five experimental replicates. The revised text is presented as follows: "Error bars represent the range between the maximum and minimum values from five independent experimental replicates, with the midpoint denoting the average value."

**Q22:** Line 316. The use of the "dual-peak signal in Bio-SPAMS" appears to contradict the statements in lines 183-184. Also, the explanation that follows is unclear. The authors appear to imply that the lower detection efficiency for 100 nm particles is simply due to a low signal-to-noise for the scattering detection…? If so, please state.

**A22:** We appreciate the reviewer's thorough evaluation. In the experimental design, a beam split mirror was employed to divide the incident beam into two parallel beams, with particle counting achieved by detecting the signal between the two beams, and the

counting results recorded on the PMT1-1 detector. This design logic is free from contradictions. Regarding the detection efficiency, a detailed explanation has been provided in Q20, and the relevant supplementary content has been integrated into the main text, significantly enhancing the logical coherence and reading fluency of the manuscript.

**Q23:** Line 321. Delete 'over'.

**A23:** We are grateful for the reviewer's recommendations. In the revised manuscript, the author has updated the relevant descriptions.

**Q24:** Fig 7. Update the y-axis to dN/dlogD (with units) as described in line 330. Also, since the test dust is non-spherical and has high density, please clarify "diameter". The direct measurements from each instrument are aerodynamic diameter in the continuum regime for the APS and aerodynamic diameter in the vacuum (or near-vacuum?) regime for Bio-SPAMS. They are related but shifted a bit from one another (eg, see DeCarlo et al., 2004). No conversion is necessary for the figure.

**A24:** Thank you to the reviewer for the suggestion. The author has modified the coordinate axis in the revised manuscript and added relevant formulas and explanations for the particle size relationship between vacuum environment and continuous flow, as follows, "The APS measures the aerodynamic diameter ($d_{ca}$) in the continuum flow regime, while the Bio-SPAMS determines the vacuum aerodynamic diameter ($d_{va}$) under vacuum or near-vacuum conditions. Since the tested samples were high-density non-spherical particles (with particle density $\rho_p > \rho_0$, where $\rho_0 = 1 \text{g/cm}^3$ is the standard reference density), the theoretical ratio between the two aerodynamic diameters can be expressed as:

$$\frac{d_{va}}{d_{ca}} = \sqrt{\frac{\rho_p \chi_c}{\rho_0 (\chi_v)^2}}$$

where $\chi_c$ and $\chi_v$ ($\geq 1$) are dynamic shape factors correcting aerodynamic drag in the continuum (viscous-dominated) and free-molecular (collision-dominated) regimes,

respectively. Since $\rho_p > \rho_0$ and $\chi_c/(\chi_v{}^2) > 1$ for such particles, $d_{va}$ exceeds $d_{ca}$, resulting in a rightward shift of the Bio-SPAMS distribution curve relative to APS in Fig. 7, where X-axis represents aerodynamic diameter."

**Q25:** Line 350. The beam width does not actually decrease. I believe the authors mean to say that "the radial distribution of particles in the buffer chamber exhibits an inverse correlation with particle size" (line 274). Reword.

**A25:** We are grateful for the reviewer's recommendations. In the revised manuscript, the author has updated the relevant descriptions.

Respond for RC2:

Q1: On line 68, please say what you mean by large particles. Larger than 5 microns? Larger than 3 microns?

**A1:** Thank you to the reviewer for your helpful suggestion. The author has made modifications to the relevant description in the revised manuscript, as follows: "most reported lenses exhibit inefficient transport of particles larger than 5 μm."

Q2: On line 95, I would suggest starting a new paragraph after "critical hole." It's the start of a new topic discussing the pre-focus design.

**A2:** New paragraphs have been added to the revised manuscript as suggested.

Q3: Fig. 1 and the text are much clearer now that all parts are labeled in Fig. 1 and consistent terminology is used in the text. However, on line 121, I'm confused by the reference to "separation cone." Is that the same thing as the virtual impactor? If so, please use that term. If not, please identify it in Fig. 1.

**A3:** Thank you for your feedback. To avoid confusion, the author revised "separation cone" to "virtual impactor" in the manuscript.

Q4: On line 132, do you mean a tapered nozzle?

**A4:** We are grateful for the reviewer's recommendations. In the revised manuscript, the author has updated the relevant descriptions.

Q5: Lines 173-174 mention different flow rates to the OPC and Bio-SPAMS but the flow rate is indicated as 0.64 lpm for both in Fig. 2. Either update the figure or update the text.

**A5:** Thank you for the valuable comments. The flow rate annotations in Fig. 2 have

been revised, and the unit (L/min) has been added to the figure caption.

Q6: Lines 212-216. Is the 110 mm downstream of the detection laser? In the paragraph describing the Bio-SPAMS (lines 176 to 188), it would be helpful to have the distance from the lens exit to the first measuring laser, to the second measuring laser and to the detection laser. That would give the 110 mm a better context.

A6: The length refers to the distance of 110 mm from the lens. To improve readability, the revised manuscript now includes the distances from the lens exit to PMT1 (42 mm) and PMT2 (67 mm). The modified text reads: "The reason for choosing the 110 mm position is that it is downstream of all lasers, including the acceleration nozzle to PMT1 (42 mm) and PMT2 (67 mm), allowing for a clearer evaluation of the particle transport and focus effect."

Q7: Lines 208 to 221, please indicate which figure the various lines are in. I would not use the word "original" on line220 because it is not referring to the line labeled original in the figure.

A7: To avoid confusion, the author has revised the corresponding text. The modified statement reads: "As shown in Fig. 3, by comparing the particle transmission efficiency curves before and after adding a virtual impactor, it can be found that after the addition of the virtual impactor (orange diamond symbol line), the focusing ability of the injection system for particles larger than 1 μm has increased to varying degrees, and the transmission efficiency of 7 μm particles has increased from the initial 5 % to 30 %."

Q8: Figure 4. Can you label the panels as a,b,c,d as you have for the other figures? I would use the same y-axis scale for all panels. It is very hard to compare when the scales are all different. For example, if all are scaled the same, then the statement on lines274-277 would be easy to evaluate. I would include the particle size in each panel.

A8: Thank you to the reviewer for your constructive comments. To avoid confusion, the author has revised the labels of the subfigures in Fig. 4. Additionally, the position of the title in Fig. 4 has been adjusted accordingly to ensure clarity and prevent overlap

with the labels in the figure.

Q9: Figure 5. I would put the particle size in each panel.

**A9:** Thank you to the reviewer for your helpful suggestions. To further enhance readability, the author has added particle size descriptions to the subfigures in Fig. 5, ensuring a clearer presentation of the relevant information.

Q10: Figure 6. I would put the numbers in a table rather than in the figure. The numbers clutter up the figure.

**A10:** To avoid confusion, the author has removed the numerical labels in Fig. 6 and added Table 1 to summarize the transmission efficiencies. Additionally, the following explanation has been included: "The trend in transmission efficiency for 10 μm particles is consistent with the simulation results, registering only 25 % efficiency.  The specific transmission efficiencies for different particle sizes are summarized in Table 1."

Table 1. Comparison of Transmission Efficiency Across Different Particle Sizes

| Particle diameter (nm) | Transmission efficiency (%) | Particle diameter (nm) | Transmission efficiency (%) |
|---|---|---|---|
| 100 | 35 | 4000 | 95 |
| 200 | 100 | 5000 | 91.65 |
| 300 | 107 | 6000 | 88.3 |
| 500 | 104 | 7000 | 72.1 |
| 800 | 101 | 8000 | 65.7 |
| 1000 | 108.5 | 9000 | 64.9 |
| 2000 | 101 | 10000 | 25 |
| 3000 | 107.5 | | |

Q11: Lines 326-330. You refer to the colors of the lines before you have referred to the figure. Please reorder these sentences.

**A11:** We appreciate the reviewer's careful reading and constructive suggestions. The

author has added a description indicating that the lines are derived from Fig. 7. Additionally, the related content on particle size distribution has been revised for greater clarity and conciseness.

Q12: Line 360. Where is the data available? Give the URL.

**A12:** The data used in the report are directly accessible, and no additional URLs are provided.

Q13: There are still a few minor issues with English language usage, such as missing words (e.g., missing "and" before "a five-stage lens" in line 16 and missing "the" before "Aerodynamic Particle Sizer" in line 28 and many other instances.) There are minor issues with citations, such as incorrect capitalization of "Mcmurry" in line 67, and inclusion of author name in the parentheses after citing the name in the text. It should just be the year in the parentheses, e.g., line 74 and many other instances.

**A13:** We are thankful to the reviewer for the valuable input. The revisions have been made in accordance with the reviewers' comments, such as including only the year after the authors' names in the references. For example:

"We typically assess the particle transmission capacity of injection systems by considering the range where the transmission efficiency exceeds 50 %, such as the 25-250 nm of Liu et al. (1995), the 100-900 nm and the 340-4000 nm of Schreiner et al. (1998; 1999), the 60-600 nm of Zhang et al. (2004), and the 125-600 nm of Zelenyuk et al. (2015). For example, Lee et al.(2013) designed a seven-stage lens for particle detection in the range of 30 nm to 10 μm, but this study does not consider the impact of critical hole on the transmission loss of large particles and it is not applied in practice. Cahill et al. (2014) designed a high-pressure lens, and used a very long buffer chamber combined with a seven-stage lens to transport 4-10 μm particles."

Other modifications have been highlighted in red in the manuscript.